



# Adjoint Optimization of Wind Plant Layouts

Ryan N. King[1,2], Katherine Dykes[2], Peter Graf[2], and Peter E. Hamlington[1]

[1]University of Colorado, Boulder, Colorado, USA
[2]National Renewable Energy Laboratory, Golden, Colorado, USA

*Correspondence to:* Ryan King (ryan.king@nrel.gov)

**Abstract.** Using adjoint optimization and three-dimensional Reynolds-averaged Navier Stokes (RANS) simulations, we present a new gradient-based approach for optimally siting wind turbines within utility-scale wind plants. By solving the adjoint equations of the flow model, the gradients needed for optimization are found at a cost that is independent of the number of control variables, thereby permitting optimization of large wind plants with many turbine locations. Moreover, compared to the common approach of superimposing prescribed wake deficits onto linearized flow models, the computational efficiency of the adjoint approach allows the use of higher-fidelity RANS flow models which can capture nonlinear turbulent flow physics within a wind plant. The RANS flow model is implemented in the Python finite element package `FEniCS` and the derivation of the adjoint equations is automated within the `dolfin-adjoint` framework. Gradient-based optimization of wind turbine locations is demonstrated on idealized test cases that reveal new optimization heuristics such as rotational symmetry, local speedups, and nonlinear wake curvature effects. Layout optimization is also demonstrated on more complex wind rose shapes, including a full annual energy production (AEP) layout optimization over 36 inflow directions and 5 windspeed bins.

## 1 Introduction

Optimizing wind turbine locations within a wind plant is a uniquely challenging problem that combines turbulent flow control with practical engineering challenges concerning the economical development of renewable energy. The problem is further complicated by the strong nonlinear coupling between turbine locations, power production, atmospheric boundary layer turbulence, and mechanical loads on turbine components. Wind plant optimization techniques used in industry often rely on heuristic guidelines and simplified linear flow models that limit computational costs. However, these approaches neglect important turbulent flow physics and can result in the underperformance of wind plants relative to their pre-construction estimates. The reduced power output and increased uncertainty due to optimization with low-fidelity flow models ultimately increases the levelized cost of energy (LCOE) and associated project risk for investors.

Major impediments to improving the LCOE for utility-scale wind plants are the difficulty of performing the turbine layout optimization using fluid dynamical models with sufficient fidelity and performing optimization in high-dimensional design spaces. In order to improve wind plant optimization, we present `WindSE`, a wind plant layout tool that uses a three-dimensional (3D) Reynolds-averaged Navier-Stokes (RANS) flow model and gradient-based optimization. This gradient-based approach, made possible by automated adjoint derivations, yields an efficient, high-fidelity, and high-dimensional wind plant optimization framework. The software is part of the National Renewable Energy Laboratory Wind-Plant Integrated System Design &



Engineering Model (WISDEM™) (Dykes et al., 2014), which provides a modular framework for comprehensive analysis and optimization of wind plant designs. Through the use of a higher-fidelity 3D RANS flow model and gradient-based adjoint optimization, `WindSE` is able to provide optimized layouts with arbitrary topological complexity at relatively low computational cost, while at the same time more accurately capturing turbulent flow effects present in real wind plants.

This paper is organized as follows: background on wind plant flow modeling, layout optimization, and adjoint optimization techniques is provided in Section II; a description of the methods used in `WindSE` for implementing the wind plant optimization problem, flow model, turbine representation, numerics, and optimization algorithm are described in Section III; a demonstration of the flow model capabilities and optimization results for idealized and real world test cases with increasingly complex wind roses, culminating in a full annual energy production (AEP) optimization, is presented in Section IV; and the paper is concluded
with a discussion of the results and implications for wind plant design in Section V.

## 2    Background

### 2.1    Wind Plant Flow Modeling

Utility-scale wind plants in the United States typically involve tens to hundreds of turbines arranged in semi-regular arrays. The layout topology is generally an outcome of optimizing the power production or net capacity factor subject to competing influences from site constraints, the local wind resource, and construction costs. These constraints include the patchwork of
viable building areas formed by leases and setbacks due to environmental concerns or physical infrastructure, terrain and soil characteristics like slope or vegetation, turbine manufacturer spacing requirements, and continuity requirements imposed by access roads and electrical connections. This results in a complex design problem, with turbine layouts varying drastically between different geographic locations and exhibiting complex topologies.

The AEP from a wind plant layout has traditionally been, and generally continues to be, assessed using reduced-order linear flow models. Such models estimate the relative wind speed across a site based on the linearized Navier-Stokes equations and treat terrain features as perturbations in boundary conditions. The underlying governing equations, introduced by Jackson and Hunt (Jackson and Hunt, 1975) and implemented in packages such as MS3DJH/3R (Walmsley et al., 1986), are based on analytical perturbation solutions to flow over a low hill. This approach decomposes the terrain into sinusoidal hills and calcu-
lates relative speedup effects over each hill. Speedup effects from multiple hills are superimposed to obtain relative velocities over the entire site. The emphasis on relative velocities is motivated by the need to extrapolate from point measurements at meteorological towers to velocity fields covering the entire plant.

A velocity deficit representing the turbine wake is then superimposed on the background wind resource at each turbine location. The PARK model developed by Jensen (Jensen, 1983) and the eddy viscosity model developed by Ainslie (Ainslie, 1988)
are commonly used, and a comprehensive review of wake modeling can be found in Crespo et al. (Crespo et al., 1999). Such approaches decouple the wind flow calculation from the wake calculation and use wake models calibrated to a single turbine in isolation. As a result, the wake model approach neglects flow curvature, speed-up effects around turbines, and changes to wakes deep in a plant. This results in known inaccuracies in complex terrain, *ad hoc* model adjustments for overlapping wakes



from multiple turbines, and systemic under-predictions of wake losses and over-predictions of power output (Barthelmie et al., 2009). Despite these limitations, linear flow models and prescribed wake models are commonly used in engineering practice because they are computationally inexpensive and reasonably accurate in simple terrain with few turbines.

Recently, higher-fidelity computational fluid dynamics (CFD) models have been increasingly applied to the study of wind plant performance. Several RANS models have been developed for simulating atmospheric flows in wind plants using actuator disk turbine representations, with a particular emphasis on modified $k-\epsilon$ closures (Cabezón et al., 2011; El Kasmi and Masson, 2008; van der Laan et al., 2015a, b). These approaches greatly improve upon linearized flow models and, given an appropriate model for the turbulent eddy viscosity, better capture turbulent transport effects in the shear layer at the edge of the wake. Large eddy simulations (LES) are also increasingly being applied to wind plant modeling (Calaf et al., 2010; Porté-Agel et al., 2011; Churchfield et al., 2012), as well as to some aspects of wind plant optimization. These simulations are vastly more expensive than RANS approaches, but resolve time-varying turbulent motions up to the filter cutoff scale. This allows for better characterization of wake meandering and unsteady loads on turbines. Finally, wind plant modeling has also been improved through coupling to numerical weather prediction simulations (Fitch et al., 2012; Mirocha et al., 2014). Such simulations are able to incorporate synoptic-scale weather forcing and interactions between the surface, wind plant, and full atmospheric boundary layer.

## 2.2 Approaches to Wind Plant Optimization

To date, wind plant layout optimization has been performed primarily using linear flow models with analytical wake deficits. The relatively cheap computational cost of such models allows gradient-free methods to be used in the optimization process. A number of layout optimization studies have utilized gradient-free approaches to minimize wake losses (Kusiak and Song, 2010), maximize net present value (González et al., 2010), and minimize noise propagation (Kwong et al., 2012). Other optimization approaches include particle swarm optimizations that determine turbine layouts and rotor diameters (Chowdhury et al., 2012), extended pattern searches for multimodal layout optimization (Du Pont and Cagan, 2012), and even game-theoretic methods (Marden et al., 2013). An exhaustive review of wind plant optimization efforts has been compiled by Herbert-Acero et al. (Herbert-Acero et al., 2014) that surveys the wide variety of objective functions, flow models, and constraints that have been studied. Results from these prior optimizations are, however, heavily influenced by the use of analytical wake models which do not fully account for nonlinear and turbulent flow physics. Consequently, significant differences in optimal layouts are expected when using higher-fidelity flow models.

Recently, higher-fidelity CFD flow models have been used in a limited range of wind plant optimization applications. The Technical University of Denmark has developed TOPFARM (Larsen et al., 2011), which employs an improved wake model (the dynamic wake meandering model) as well as a parabolic Navier-Stokes solver. TOPFARM uses a hybrid optimization approach that combines sequential linear programming (SLP) with gradient-free genetic algorithms. The genetic algorithm is used to find the neighborhood of the global optimum, and then the gradient-based SLP algorithm completes the optimization. However, the genetic algorithm step penalizes large design spaces, limiting TOPFARM to relatively small wind plants. King et al. used adjoints of a 2D RANS flow model to optimize turbine locations and observed that the 2D nature of the flow



solver resulted in substantial flow curvature (King et al., 2016). Meyers and Meneveau (Meyers and Meneveau, 2012) studied the effects of spacing and alignment in infinite wind plants with turbines arranged in regular gridded turbine arrays. In their study, pseudospectral LES was used to model atmospheric boundary layer turbulence and complex wake interactions, but the optimized layouts were restricted to grids, as opposed to a more general layout topology optimization where turbines are

free to arrange in non-gridded configurations. LES results have also been used to tune linear flow models that were used in optimization of yaw control (Gebraad et al., 2016), turbine layouts (Bokharaie et al., 2016), and coupled layout and yaw optimization (Fleming et al., 2016), or in the tuning of RANS models for wind plant control optimization (Iungo et al., 2016). Wind plant LES simulations have also been used to directly perform adjoint optimization of wind plant controls by adjusting rotor thrust during operation (Goit and Meyers, 2015; Goit et al., 2016). These approaches benefit from high-fidelity CFD

and leverage the power of adjoint optimization, but only considered fixed layouts. Finally, Funke et al. (Funke et al., 2014) performed an adjoint optimization of ocean turbine layouts using an analysis based on the shallow water equations, with ocean turbines represented by increased bottom friction.

In the present study, we use adjoint techniques to enable gradient-based optimization of wind turbine locations within a plant, subject to realistic turbulent flow fields. A 3D RANS flow solver is employed as a first-principles model that can predict

new turbulent flow physics, rather than prescribing fixed wake behaviors as in linear flow models. The RANS model provides an accurate model of a neutral atmospheric boundary layer at moderate computational cost and without requiring calibration using LES. The RANS model is also amenable to automatic differentiation and gradient-based optimization, as explained in the next section. This gradient-based approach permits the use of high-dimensional control spaces, thereby providing optimized layouts of arbitrary complexity (i.e., optimized layouts are not restricted to grids or any other regular arrangement). We further

demonstrate layout optimization for the full plant AEP based on a real site wind rose, going beyond the uniform speed optimization considered previously (King et al., 2016). The resulting optimization framework thus represents a novel application of adjoint techniques to the optimization of utility-scale wind plants using a higher-fidelity CFD flow model.

## 2.3  Adjoint Techniques for Efficiently Calculating Gradients

The greater expense of CFD flow models such as RANS and LES compared to linearized flow models requires the use of an

efficient optimization technique that minimizes the number of flow model evaluations. Gradient-based optimization methods are promising candidates for CFD-driven wind plant optimization since they require orders of magnitude fewer function evaluations than gradient-free techniques. However, finding these gradients can be challenging when using complex flow models or when there are many control variables. Calculating such gradients with a finite-difference approach requires a function evaluation for each control variable, making this approach prohibitively expensive for optimizing utility-scale plants with many

turbines.

In the present optimization framework, the necessary gradients are obtained relatively inexpensively by using adjoint optimization techniques. A comprehensive review of discrete techniques for calculating gradients of engineering design problems, including the adjoint approach, can be found in Martins and Hwang (Martins and Hwang, 2013). The adjoint approach allows one to calculate gradients at a cost that scales with the dimension of the objective function rather than with the number of





design variables. For wind plant optimizations with scalar objective functions, this means that gradients can be found at a fixed cost regardless of the number of control variables.

Adjoint systems arise naturally from consideration of dynamical systems whose behavior can be described by differential equations, and they have a rich history in fluid dynamics and sensitivity analysis. Reviews of adjoint techniques in various

flow control and optimization applications have been published by Jameson (Jameson, 2003), Giles and Pierce (Giles and Pierce, 2000), Giannakoglou and Papadimitriou (Giannakoglou and Papadimitriou, 2008), and Luchini and Bottaro (Luchini and Bottaro, 2014). At its core, the adjoint approach reverses the propagation of information in a dynamical system. Given an ordinary or partial differential equation governing the evolution of a dynamical system and an objective function measuring a quantity of interest, the adjoint approach produces a differential equation whose states evolve backward in time and indicates

where a perturbation would maximally influence the objective function. By reversing the flow of information, the adjoint reveals what is effectively the optimal open-loop control input.

The adjoint operator is defined by the bilinear identity $\langle Au, v \rangle = \langle u, Bv \rangle$, where operator $B$ is adjoint to operator $A$. This holds if $A$ is a continuous differential operator, in which case $B$ is found through integration by parts, or if $A$ is a matrix, in which case $B = A^*$, where $A^*$ is the complex conjugate transpose of $A$. This bilinear identity reveals that the adjoint operator

is implicitly defined by the forward model, but in order for the adjoint problem to be well posed, additional constraints are required. Because the adjoint travels backwards in time, these constraints are terminal conditions rather than initial conditions, and they come from the specific objective function under consideration as well as from the states produced by the forward model.

The following general framework illustrates the computational advantages of the adjoint approach. Consider a dynamical

system with governing equations that can be expressed in residual form as $\mathcal{F}(\mathbf{u}, \mathbf{m}) \equiv 0$, where $\mathcal{F}$ is a vector-valued differential equation (e.g., the RANS equations), $\mathbf{u} \in \mathbb{R}^n$ is the system state vector (e.g., the flow field velocities), and $\mathbf{m} \in \mathbb{R}^m$ is a vector of control variables (e.g., the wind turbine coordinates). Additionally, consider a scalar objective functional $J(\mathbf{u}, \mathbf{m}) \in \mathbb{R}$ that measures a quantity of interest (e.g., the LCOE). Many engineering problems can be formulated as constrained optimization problems that seek optimal control parameters to minimize $J$, namely

$$\min_{\mathbf{m}} \quad J(\mathbf{u}, \mathbf{m}) \tag{1}$$

$$\text{subject to} \quad \mathcal{F}(\mathbf{u}, \mathbf{m}) = 0 \tag{2}$$

$$h(\mathbf{m}) = 0 \tag{3}$$

$$g(\mathbf{m}) \leq 0 \tag{4}$$

where $h$ and $g$ are additional equality and inequality constraints on the control parameter $\mathbf{m}$, such as upper and lower bounds on

a control input (e.g., wind plant site boundaries). In common engineering problems, $\mathcal{F}$ is a partial differential equation (PDE) that is expensive to evaluate, and the dimensions of both the control space and state space are high. Efficiently solving this PDE-constrained optimization problem requires algorithms that scale well to high dimensions and that minimize the number of evaluations of $\mathcal{F}$.





Gradient-based optimization algorithms can approach second-order convergence to local minima and can minimize the evaluations of $\mathcal{F}$, but require the gradient of the objective functional with respect to all of the control parameters. This gradient, $\mathrm{d}J/\mathrm{d}\mathbf{m}$, is given by the chain rule as

$$\frac{\mathrm{d}J}{\mathrm{d}\mathbf{m}} = \frac{\partial J}{\partial \mathbf{m}} + \frac{\partial J}{\partial \mathbf{u}}\frac{\partial \mathbf{u}}{\partial \mathbf{m}}. \tag{5}$$

Since $J(\mathbf{u},\mathbf{m})$ is generally a user-defined function, $\partial J/\partial \mathbf{m} \in \mathbb{R}^{1\times m}$ and $\partial J/\partial \mathbf{u} \in \mathbb{R}^{1\times n}$ are straightforward to determine analytically. However, $\partial \mathbf{u}/\partial \mathbf{m} \in \mathbb{R}^{n\times m}$ is expensive to compute for high-dimensional control and state spaces. A finite difference approach to calculating this gradient would require $m$ evaluations of $\mathcal{F}$, which is intractable for many engineering problems.

In the adjoint approach, $\partial \mathbf{u}/\partial \mathbf{m}$ is eliminated from Eq. (5) by taking the derivative of the PDE constraint $\mathcal{F}(\mathbf{u},\mathbf{m}) = 0$ with
respect to $\mathbf{m}$, resulting in the tangent linear system

$$\frac{\partial \mathcal{F}}{\partial \mathbf{m}} + \frac{\partial \mathcal{F}}{\partial \mathbf{u}}\frac{\partial \mathbf{u}}{\partial \mathbf{m}} = 0. \tag{6}$$

Solving for $\partial \mathbf{u}/\partial \mathbf{m}$ in Eq. (6) and substituting into Eq. (5) then yields

$$\frac{\mathrm{d}J}{\mathrm{d}\mathbf{m}} = \frac{\partial J}{\partial \mathbf{m}} - \underbrace{\frac{\partial J}{\partial \mathbf{u}}\left[\frac{\partial \mathcal{F}}{\partial \mathbf{u}}\right]^{-1}}_{\boldsymbol{\Psi}^{\mathsf{T}}}\frac{\partial \mathcal{F}}{\partial \mathbf{m}}, \tag{7}$$

where we have introduced the new *adjoint* variable $\boldsymbol{\Psi}$. This variable maps source perturbations in $\mathcal{F}(\mathbf{u},\mathbf{m}) = 0$ to sensitivities
of $J$. By definition, it is governed by

$$\left[\frac{\partial \mathcal{F}}{\partial \mathbf{u}}\right]^{\mathsf{T}}\boldsymbol{\Psi} = \left[\frac{\partial J}{\partial \mathbf{u}}\right]^{\mathsf{T}}. \tag{8}$$

With a solution of the forward model $\mathcal{F}$ and the adjoint $\boldsymbol{\Psi}$, the derivative of the objective function expressed in Eq. (5) can then be calculated as

$$\frac{\mathrm{d}J}{\mathrm{d}\mathbf{m}} = \frac{\partial J}{\partial \mathbf{m}} - \boldsymbol{\Psi}^{\mathsf{T}}\frac{\partial \mathcal{F}}{\partial \mathbf{m}}. \tag{9}$$

The resulting adjoint gradients are typically more accurate than finite difference gradients, and because Eq. (8) is independent of $\mathbf{m}$, the gradient can be calculated at a fixed cost regardless of the dimension of $\mathbf{m}$. This enables efficient gradient-based optimization for systems governed by computationally-demanding PDEs. Equation (8) provides an intuitive interpretation of the adjoint as a linear mapping between source perturbation in the governing equations and changes in the objective function. The adjoint thus provides the optimal control input and directly encodes the sensitivities of $J$. Alternatively, the adjoint can
also be interpreted as a Lagrange multiplier enforcing the PDE constraint $\mathcal{F}$.

## 3 Methodology

In the present study, wind plant layout optimization is approached as a PDE-constrained optimization problem using the adjoint theory developed in the previous section. The wind plant power output is maximized with gradient-based optimization



techniques, subject to a PDE constraint corresponding to the RANS equations, which are used to predict turbulent flow within the plant.

## 3.1 Optimization Problem Definition

We seek to maximize steady-state power output from $N$ different turbines experiencing $K$ different inflow wind states (each state is defined by a wind speed and direction) by controlling the 2D Cartesian positions of the turbines, $\mathbf{x} = (x_1 \ldots x_N)$ and $\mathbf{y} = (y_1 \ldots y_N)$. The Reynolds-averaged velocity field $\overline{u}_i \in \mathbb{R}^3$ and pressure field $\overline{p} \in \mathbb{R}$ are taken to be the states (denoted $\mathbf{u}$ in the preceding discussion of adjoints) and the turbine coordinates are the control variable $\mathbf{m} = [\mathbf{x}, \mathbf{y}]^\mathsf{T} \in \mathbb{R}^{2N}$. This leads to the following optimization problem:

$$\min_{\mathbf{m}} \quad J = -\sum_{k=1}^{K} \sum_{n=1}^{N} \alpha_k \frac{1}{2} \rho A C'_{p,n} \|\overline{\mathbf{u}} \cdot \hat{\mathbf{n}}_k\|^3 \tag{10}$$

$$\text{subject to} \quad \overline{u}_j \frac{\partial \overline{u}_i}{\partial x_j} = -\frac{1}{\rho} \frac{\partial \overline{p}}{\partial x_i} + \nu \frac{\partial^2 \overline{u}_i}{\partial x_j^2} - \frac{\partial \tau_{ij}}{\partial x_j} + \frac{1}{\rho} \sum_{n=1}^{N} f_{AD,n} \hat{\mathbf{n}}_k \tag{11}$$

$$\frac{\partial \overline{u}_i}{\partial x_i} = 0 \tag{12}$$

$$f_{AD,n} = \frac{1}{2} \rho A C'_{t,n} \|\mathbf{u} \cdot \hat{\mathbf{n}}_k\|^2 \tag{13}$$

$$\tau_{ij} = -\nu_T \overline{S}_{ij} \tag{14}$$

$$\nu_T = \ell_{mix}^2 \left( 2 \overline{S}_{ij} \overline{S}_{ij} \right)^{1/2} \tag{15}$$

$$L_{x,lower} < x_n < L_{x,upper} \tag{16}$$

$$L_{y,lower} < y_n < L_{y,upper} \tag{17}$$

$$(x_i - x_j)^2 + (y_i - y_j)^2 > D_{min}^2 \quad \forall i, j = 1 \ldots N, \, i \neq j \tag{18}$$

where $J$ is the negative of the total power production, $\alpha_k$ are weights for each inflow state from the wind speed distribution and wind rose describing the site climatology, $A$ is the turbine rotor area, and $\rho$ is the air density. The PDE constraints are the steady 3D RANS and continuity equations which govern the evolution of $\overline{u}_i$ and $\overline{p}$. The RANS equations have an additional body force term $f_{AD,n}$ that represents the force imparted on the flow by each wind turbine. The Boussinesq hypothesis is used to close the deviatoric Reynolds stress term $\tau_{ij}$ by making use of an eddy viscosity $\nu_T$. The eddy viscosity is calculated with a mixing length model (Wilcox, 2006), where the mixing length $\ell_{mix}$ is taken to be $1/8^{\text{th}}$ the vertical distance from the bottom surface. The power and thrust coefficients for turbine $n$, denoted $C'_{p,n}$ and $C'_{t,n}$, respectively, are derived from actuator disk theory and are described in the next section. The site constraints are imposed as lower, $L_{x,lower}$ and $L_{y,lower}$, and upper, $L_{x,upper}$ and $L_{y,upper}$, bounds on the turbine $\mathbf{x}$ and $\mathbf{y}$ locations. An inter-turbine minimum spacing constraint is further enforced with a minimum spacing $D_{min}$ of three times the rotor diameter (RD) used in the AEP simulations. The turbines are assumed to always yaw into the wind and the turbine body force is thus directed into the incoming wind by the unit vector $\hat{\mathbf{n}}_k$. Note that





the subscript $n$ in the above system of equations represents one of the $N$ different turbines and the subscript $k$ denotes one of the $K$ different wind states included in the analysis.

We stress that the strength of this approach is not the particular form of the RANS closure model, but rather its embedding in an adjoint optimization framework. Different objective functions, turbulence closures, or turbine representations can be easily implemented in this framework and still benefit from the adjoint approach.

## 3.2 Wind Turbine Representation

Turbines in the simulations are represented as non-rotating actuator disks using actuator disk theory, as described in standard wind energy texts (Burton et al., 2011). The power, $P$, and thrust force, $T$, generated by an actuator disk are given in terms of a power coefficient, $c_p$, a thrust coefficient, $c_t$, and an upstream reference velocity, $u_\text{ref}$, as

$$P = \frac{1}{2}\rho A c_p u_\text{ref}^3, \quad T = \frac{1}{2}\rho A c_t u_\text{ref}^2, \tag{19}$$

where power and thrust coefficients are given in terms of an axial induction factor $a$ as

$$c_p = 4a\left(1-a\right)^2, \quad c_t = 4a\left(1-a\right). \tag{20}$$

The axial induction factor describes the velocity at the rotor disk, $u_\text{rotor} = u_\text{ref}\left(1-a\right)$, as a fraction of the upstream reference velocity. The reference velocity is typically taken to be the far-field upstream velocity for a turbine in isolation, but as discussed by Sanderse et al. (Sanderse et al., 2011), the determination of $u_\text{ref}$ for waked turbines or in complex terrain is more difficult. We adopt the same approach used by Calaf et al. (Calaf et al., 2010) and Meyers and Meneveau (Meyers and Meneveau, 2012) and define modified power and thrust coefficients, $c_p'$ and $c_t'$, respectively, that are based on the local rotor disk velocity $u_\text{rotor}$ rather than $u_\text{ref}$. These modified coefficients are

$$c_p' = \frac{c_p}{\left(1-a\right)^3}, \quad c_t' = \frac{c_t}{\left(1-a\right)^2}, \tag{21}$$

resulting in power and thrust given by

$$P = \frac{1}{2}\rho A c_p' u_\text{rotor}^3, \quad T = \frac{1}{2}\rho A c_t' u_\text{rotor}^2. \tag{22}$$

For this study, turbine operating parameters of $c_t = 3/4$, $c_p = 0.34$, and $a = 1/4$ are used for all turbines, consistent with values used in previous CFD studies (Calaf et al., 2010; Meyers and Meneveau, 2012; Jimenez et al., 2007). This results in modified power and thrust coefficients of $c_p' = 0.81$ and $c_t' = 4/3$.

Standard actuator disk theory assumes that the rotor disk is uniformly loaded, but this introduces singularities at the rotor edge. To ensure that the thrust force and power production are continuously differentiable, as well as to avoid numerical instabilities, the turbine force and power production are smoothly distributed across the rotor swept area. This differentiability with respect to position is crucial for gradient-based layout optimization, and smoothly distributing rotor forces is common practice in actuator disk (Wu and Porté-Agel, 2010) and actuator line (Churchfield et al., 2012) implementations. Effectively, this smoothing transforms the scalar $c_p'$ and $c_t'$ coefficients into continuously differentiable fields $C_{p,n}'\left(x,y,z\right)$ and



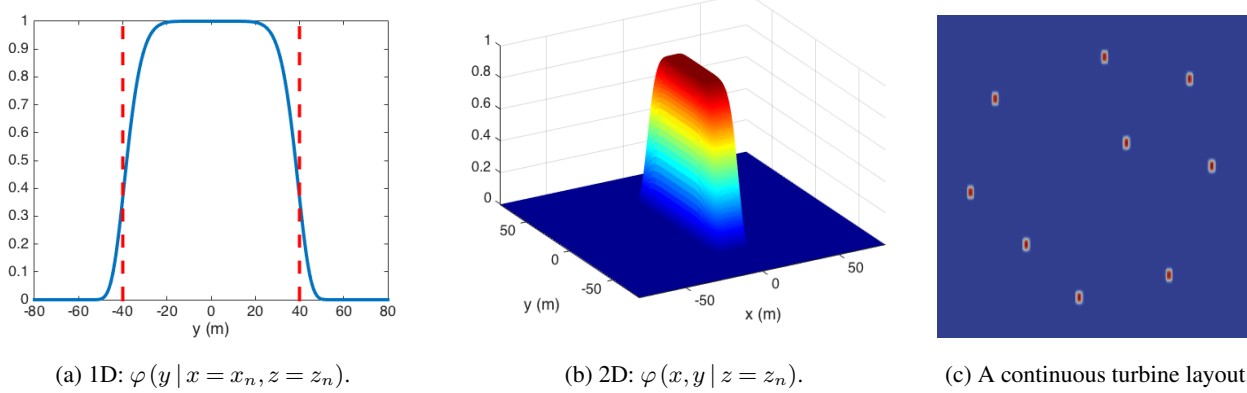

(a) 1D: $\varphi\left(y \mid x=x_n, z=z_n\right)$.      (b) 2D: $\varphi\left(x, y \mid z=z_n\right)$.      (c) A continuous turbine layout.

**Figure 1.** A continuously differentiable modified exponential distribution is used to smoothly distribute the turbine forces and power production over the rotor swept area. The smoothing function is demonstrated for an 80 m rotor diameter turbine with $\gamma = 6$ for 1D and 2D prototypes in panel (a) and (b) by taking conditional distributions of $\varphi(x, y, z)$ in Eq. (23). The edge of the rotor disk is represented by vertical dashed red lines in panel (a). Panel (c) is a hub-height slice showing how a wind plant is represented as a summation of $C'_{t,n}(x, y, z)$ fields from each turbine.

$C'_{t,n}(x, y, z)$, parameterized by the location of a particular turbine $(x_n, y_n)$, the turbine hub height $z_n$, and a smoothing kernel $\varphi(x, y, z ; x_n, y_n, z_n)$. The following multivariate exponential distribution is used as the smoothing kernel in order to create a rotor with almost compact support that is still continuously differentiable:

$$\varphi\left(x, y, z ; x_n, y_n, z_n\right) = \exp\left\{-\left[\frac{x-x_n}{w}\right]^\gamma - \left[\frac{(y-y_n)^2 + (z-z_n)^2}{r^2}\right]^\gamma\right\}, \tag{23}$$

5  where $\varphi(x, y, z ; x_n, y_n, z_n)$ is the distribution for a single turbine centered on $(x_n, y_n, z_n)$, $r$ is the rotor radius, $w$ is the rotor halfwidth, and $\gamma$ controls the sharpness of the rotor edge. Both 1D and 2D prototypes of this smoothing function are shown in Figs. 1(a) and (b) for $\gamma = 6$ and $w = r/4$, with $r = 40$ m.

The smoothing kernel in Eq. (23) is used to express smooth power and thrust coefficient fields, $C'_{p,n}(x, y, z)$ and $C'_{t,n}(x, y, z)$, respectively, for each turbine as

10  $$C'_{p,n}(x, y, z) = \beta^{-1}\varphi(x, y, z ; x_n, y_n, z_n)c'_p, \quad C'_{t,n}(x, y, z) = \beta^{-1}\varphi(x, y, z ; x_n, y_n, z_n)c'_t, \tag{24}$$

where a normalization constant, $\beta$, is introduced so that integration of $C'_{p,n}(x, y, z)$ and $C'_{t,n}(x, y, z)$ over the rotor swept volume returns the correct scalar values $c'_{p,n}$ and $c'_{t,n}$. This constant is given by

$$\beta = \int_{-\infty}^{\infty}\int_{-\infty}^{\infty}\int_{-\infty}^{\infty} \varphi(x, y, z ; x_n, y_n, z_n)\,\mathrm{d}x\mathrm{d}y\mathrm{d}z. \tag{25}$$

The continuous thrust coefficient field $C'_{t,n}(x, y, z)$ for an example turbine layout is shown with a hub-height slice in Fig. 1(c).





### 3.3 Simulation Setup

The 3D computational domain used in the simulations has horizontal dimensions of 2.4 km $\times$ 2.4 km and a vertical dimension of 640 m (equivalent to $30 \times 30 \times 8$ RD, with RD = 80 m), as shown in Fig. 2. Dirichlet boundary conditions are used to prescribe the wind speed and direction on inflow, top, and bottom boundaries, corresponding to the planes at $x = -1.2$ km, $z = 640$ m, and $z = z_0$, respectively. The inflow velocity profile is specified according to a neutral logarithmic velocity profile

$$u_{in}\left(z\right) = \frac{u_*}{\kappa} \ln\left(\frac{z}{z_0}\right), \tag{26}$$

where $z$ is the vertical coordinate, $z_0$ is the roughness height, $u_*$ is the friction velocity, and $\kappa = 0.4$ is the Von Karman constant. The roughness height is taken to be $z_0 = 0.04$ m, corresponding to open and relatively smooth terrain, and $u_*$ is found by solving for the desired hub-height velocity. A no-slip boundary condition is used on the bottom boundary and the velocity along the top boundary is $u_{in}(640 \text{ m})$ from Eq. (26). On outflow boundaries (i.e., the planes at $x = 1.2$ km, $y = -1.2$ km, and $y = 1.2$ km), a standard 'do-nothing' finite element outflow boundary condition is applied. The initial velocities at all

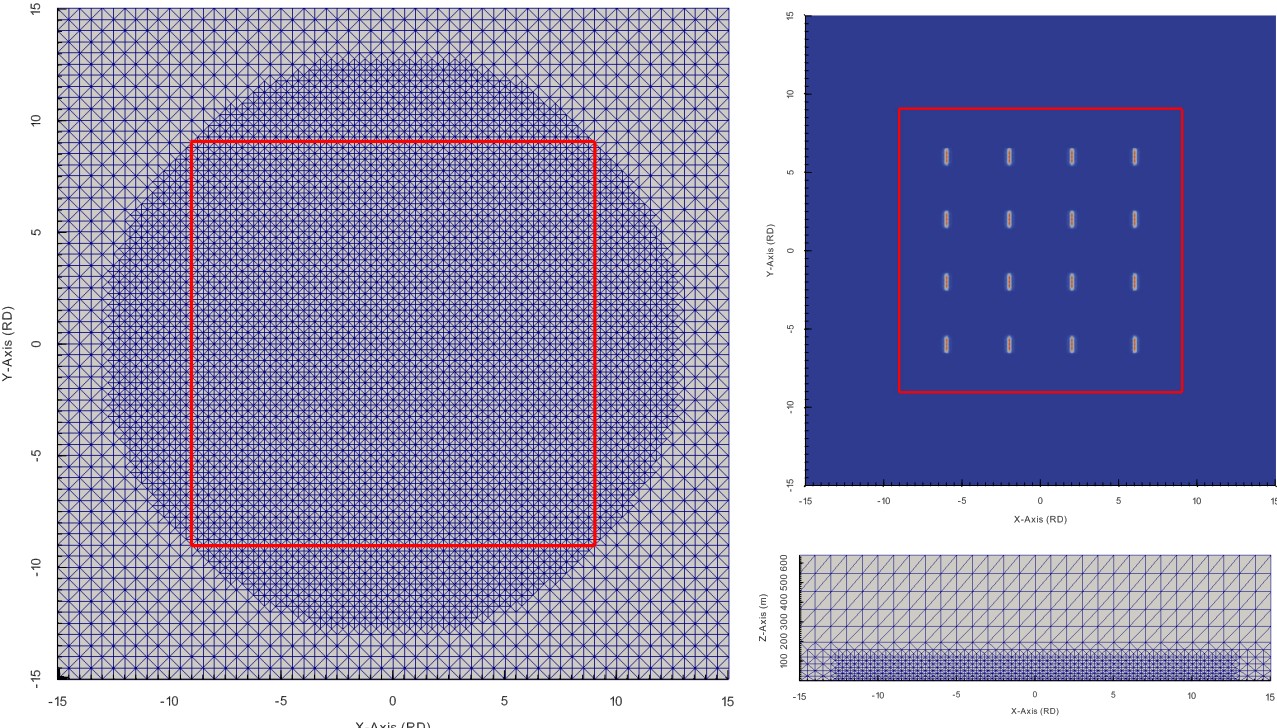

**Figure 2.** Plan view of the computational mesh at hub height showing additional refinement around the wind turbine area (left), schematic of the computational domain showing the turbine locations used to initialize each optimization (top right), and side view of the mesh showing refinement below 2 rotor diameters (bottom right). The wind plant site constraint is shown as a red square in the plan view, and horizontal dimensions are normalized by the rotor diameter RD=80 m.



locations in the domain are given by the logarithmic profile in Eq. (26), and the initial relative pressure is assumed to be 0 bar everywhere.

A coarse computational mesh of finite elements is generated for the entire domain and then further refined within a circle that circumscribes the site boundaries, as shown in Fig. 2. For all tests presented here, the site boundary is assumed to be a square of side 18 RD centered in the computational domain. This area corresponds to a 6 RD spacing in streamwise and spanwise directions if 16 turbines are arranged in a regular grid. The resolution of the finite elements used in the simulation can be quantified by the radius of a circle that circumscribes a single tetrahedral element, termed the circumradius. The mesh is refined such that the turbine rotor diameter is at least 4 times larger than the circumradius of the finite elements within the wind plant site boundaries. The mesh is also stretched by a factor of 1.2 in the vertical direction in order to increase resolution near the bottom boundary. This results in a mesh with approximately 200,000 degrees of freedom.

Simulations are performed for a range of different inflow directions and inflow speeds, corresponding to both idealized and real-world wind roses and wind speed distributions. Steady-state solutions are found for each of the $K$ wind states. We assume that the turbines are always yawed into the wind such that the rotor disk is perpendicular to the inflow direction. The computational domain and boundary conditions are assumed constant and different wind directions are modeled by applying a rotation to the turbine coordinates $\mathbf{x}$ and $\mathbf{y}$ corresponding to the inflow angle. That is, the inflow is always assumed perpendicular to the plane at $x = -1.2$ m and different inflow directions are modeled by rotating the wind plant layout about the center of the computational domain. A weighted sum of the power production for each wind state (i.e., speed and direction) is performed based on the site wind speed distribution and wind rose, and the adjoint gradient is calculated over the total power output, taking into account the layout rotation. Because the boundary conditions are part of a well-posed PDE used as an optimization constraint, this approach of rotating the layout is preferred to explicitly changing the inflow direction because the PDE constraint in the optimization is kept constant.

### 3.4 Gradient-Based Layout Optimization Process

Figure 3 shows a schematic of the multi-state optimization workflow. The layout is optimized over all inflow states simultaneously in a multilevel optimization process with respect to the total power output rather than in a sequential optimization process over each inflow separately. The basic optimization process is the following:

1. Begin with an initial layout $\mathbf{m}_i$, which can be either a gridded or random layout of the $N$ turbines.

2. Perform flow-field simulations for each of the $K$ desired wind states (corresponding to different wind speeds and directions obtained from either real or idealized wind roses and speed distributions).

3. Calculate the negative of the wind plant power, $J_k$, for each of the $K$ wind states.

4. Calculate the objective function by taking a weighted sum $J = \sum \alpha_k J_k$ over all $K$ wind states, where $\alpha_k$ is the relative probability of each wind state obtained from either a real or idealized wind rose and speed distribution.

5. Compute adjoint simulations for the forward simulations, and calculate the total objective function Jacobian d$J$/d$\mathbf{m}$.





6. Use the gradient to perform a gradient-based optimization of the layout to obtain $\mathbf{m}_{i+1}$ and go to step 2 above.

This process is repeated until the change in the objective function or its gradient falls below a user defined threshold. The optimization typically requires less than 50 iterations to converge to an optimal layout for a single wind direction, and fewer iterations when optimizing over multiple inflow states. Because the line search method used to determine step size at each iteration often requires several function evaluations, there can be more function evaluations than optimization iterations.

It should be noted that, because the optimization algorithm is gradient-based, it finds local rather than global minima in the total objective function $J$. The layout optimization problem can have many local minima, particularly with just a few inflow wind states. Starting with a regular gridded layout slightly smaller than the site constraint was generally observed to produce reasonable results since none of the variables are initially constrained and a gridded layout is often a "good enough" guess to be in the radius of convergence to the global optimum. This initial layout, used for all simulations described in the following, is shown in Fig. 2. However, gradient-based methods are still fundamentally a local search and cannot provide strong assurances of finding a global minima. We note that with many inflow states (i.e., for large $K$), the optimization problem actually becomes more convex and convergence is achieved in fewer iterations. Additionally, running the optimization from many different starting configurations generated by random sampling or Latin hypercube samples can be used to further characterize the robustness of the minima.

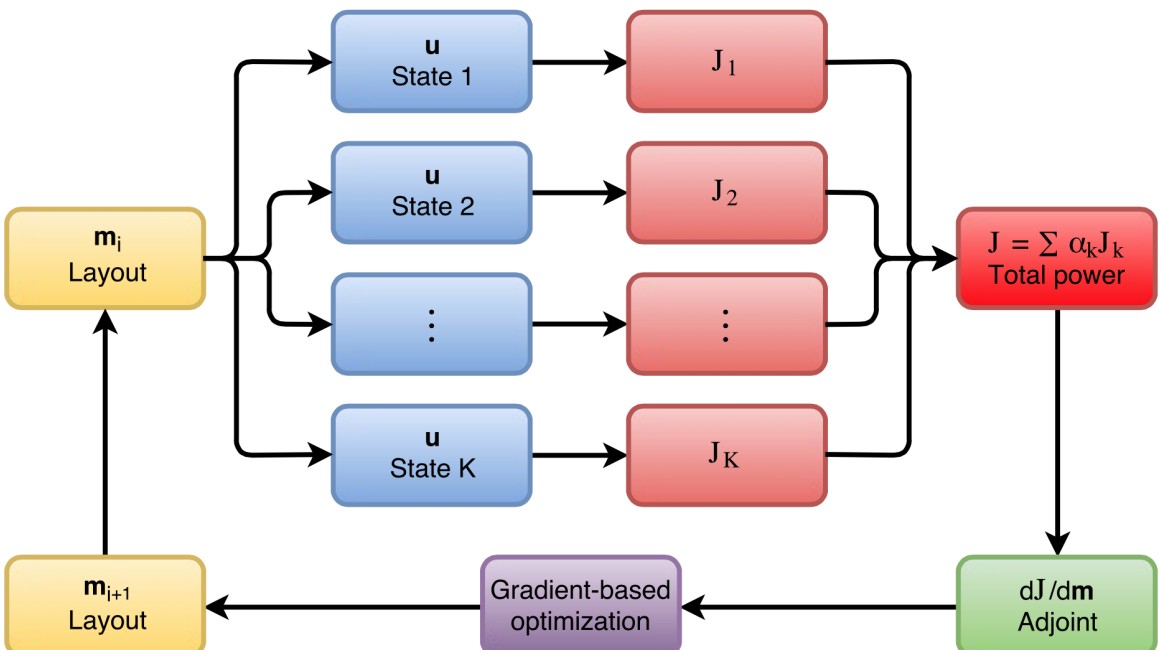

**Figure 3.** Schematic of the multilevel optimization process for a wind plant with $K$ wind states (i.e., wind speeds and directions).



### 3.5  Numerical Implementation

The WindSE flow solver is implemented in a software package called `FEniCS` (Logg et al., 2012), which automates the solution of PDEs using the finite element method. `FEniCS` is written in Python and can be easily integrated with other Python-based systems engineering tools like WISDEM. The `FEniCS` project is based on the `DOLFIN` problem-solving environment and connects a number of useful components for the automatic discretization and solution of finite element problems. These components include a form language that allows users to specify equations in variational form using a syntax that closely resembles their mathematical description, automated compilers that generate finite element forms for a chosen basis, and just-in-time compilation to C++ to enhance computational speed. `FEniCS` can interface to common HPC libraries such as PETSc and Trilinos for numerical linear algebra, ParMETIS and SCOTCH for domain decomposition, and MPI and OpenMP for parallelization. `FEniCS` has been extensively tested and validated on a number of computational problems in solid and fluid mechanics, eigenvalue problems, and coupled PDEs (Logg et al., 2012).

The description of finite element problems as variational forms in `FEniCS` lends itself to highly abstracted algorithmic differentiation. The software package `dolfin-adjoint` (Farrell et al., 2013) performs a high-level algorithmic differentiation of a forward problem implemented in `FEniCS` and can derive both discrete adjoint and tangent linear models. The discrete adjoint and tangent linear models are important in the gradient-based optimization algorithms that are used in data assimilation, optimal control, and error estimation. The algorithmic differentiation routines in `dolfin-adjoint` act on the forward problem discrete equations described in variational form and stored in memory. The corresponding discrete adjoint equations are derived in `dolfin-adjoint` and passed back to `FEniCS` as an additional PDE problem. This approach operates at a higher level of abstraction than traditional algorithmic differentiation, which typically treats forward models as a series of elementary instructions involving native operations in low-level code like addition and multiplication. This higher level of abstraction gives `dolfin-adjoint` greater flexibility and automation across a wide range of PDE applications because it avoids differentiating across low-level code where the mathematical and implementation details have been intermixed. Moreover, `dolfin-adjoint` can be implemented on unsteady and nonlinear PDEs, and can also be run in parallel. It can directly interface to the optimization algorithms in SciPy and also contains routines for checking the correctness of adjoint gradients and checkpointing.

Using `FEniCS`, Eqs. (11) and (12) are solved with a nonlinear Newton solver in a coupled fashion using a mixed finite element space with piecewise linear elements for both the velocity and pressure fields. To satisfy the Ladyzhenskaya-Babuška-Brezzi (LBB) (or inf-sup) compatibility condition (Brezzi and Fortin, 1991) with equal-order basis functions, we augment the momentum equation with an additional pressure-stabilized Petrov-Galerkin term that weights the residual of the momentum equation by the gradient of the pressure test function. This pressure-based stabilization alleviates the saddle-point nature of the equal order finite element problem (Donéa and Huerta, 2003) but still vanishes for the exact solution to the momentum equation. Each nonlinear solve is initialized with the base logarithmic velocity profile and the relative residual is converged to below $10^{-7}$ with Newton's method. The Newton solver uses Jacobians derived automatically within `FEniCS` and linear systems are solved directly with the sparse, parallel solver MUMPS (Amestoy et al., 2000). The choice of equal order piecewise linear mixed finite





element spaces differs from previous studies on wind and ocean turbine layout optimization in `FEniCS` (King et al., 2016; Funke et al., 2014) that used Taylor-Hood mixed finite element spaces that are piecewise quadratic for the velocity field. The lower order representation in this study was necessary when implementing a 3D solver to keep the total degress of freedom sufficiently low so that a direct linear algebra solver could be used.

The gradients obtained from `dolfin-adjoint` are used to optimize turbine locations with Python's SciPy implementation of the sequential least squares programming (SLSQP) algorithm. SLSQP is a gradient-based optimization algorithm that can also handle constraints (Nocedal and Wright, 1999). SLSQP minimizes a quadratic approximation to the objective function at each optimization iteration, with a linear approximation of the constraints. Gradients of the objective function are provided by `dolfin-adjoint` and gradients of the minimum turbine spacing constraint are provided manually to SLSQP.

The forward and adjoint problems are parallelized with MPI and can be run on a desktop or in a high performance computing environment. The discrete adjoint calculation is automatically parallelized by `dolfin-adjoint` if the forward model is run in parallel, which drastically simplifies code development.

## 4    Results

In the following, we present results for several different layout optimization cases. First, we provide results for standard test
cases of flow past a single turbine and flow through a very deep wind plant in order to demonstrate that the RANS flow solver accurately captures wind turbine wakes, thereby providing confidence that subsequent layout optimizations are performed according to the correct flow physics. Second, we optimize a 16-turbine wind plant using wind roses with evenly-weighted wind directions and a constant wind speed of 8 m/s in order to demonstrate new layout insights and optimization heuristics when accounting for nonlinear flow effects with a high-fidelity model. Third, we optimize a 16-turbine wind plant using
unevenly-weighted wind roses that exhibit complex directional preferences, again with a constant wind speed of 8 m/s. Finally, we perform a full AEP optimization using data from the M2 meteorological tower at the National Wind Technology Center (NWTC) to demonstrate the capabilities of `WindSE` when optimizing layouts for conditions that are representative of real wind plants. In the full AEP optimization, both the wind direction and wind speed are varied, resulting in approximately one hundred total wind states over which the optimization is performed. In all cases, we assume a turbine with a rotor diameter of
RD = 80 m operating with fixed power and thrust coefficients (as described in Section 3.2) and a site constraint that is a square of side 18 RD centered in the computational domain.

### 4.1   RANS Model Testing

As a test of the physical accuracy of the RANS flow solver, `WindSE` was used to simulate flow fields for both a single turbine and a deep wind plant. Figure 4 shows vertical and horizontal velocity profiles in the wake of a single wind turbine. Consistent
with theoretical expectations and previous results, a logarithmic velocity profile is observed in the undisturbed flow upstream of the turbine, a Gaussian velocity deficit is observed in the turbine wake, and a gradual wake recovery is observed with increasing distance downstream from the turbine. Moreover, a slight speedup around the edges of the wake is observed near the turbine





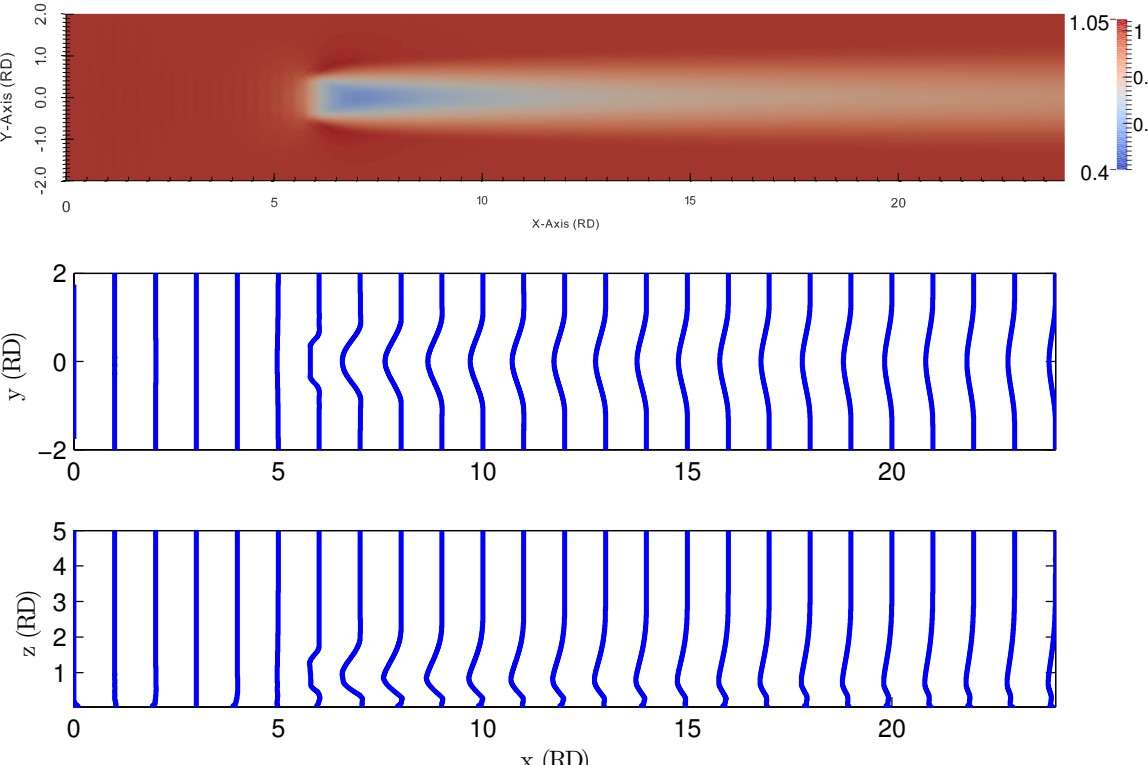

**Figure 4.** Flow past a single turbine obtained using the RANS flow solver. The top panel shows the velocity at hub height and the bottom two panels show profiles of the velocity deficit in horizontal and vertical planes passing through the center of the wind turbine rotor. Velocity deficits are relative to the respective profiles 3 rotor diameters upstream of the turbine and the velocities are normalized by the incoming hub-height velocity. Axes are in units of rotor diameter RD = 80 m.

rotor – such speedups are not captured by traditional linear wake flow models and appear here due to the use of the higher-fidelity RANS flow solver, which more accurately captures nonlinear flow physics. Figure 5 shows the velocity field from a simulation of a wind plant with 10 turbine rows perpendicular to the incoming wind direction. This flow field shows good qualitative agreement with time-averaged LES results reported by Wu and Porté-Agel (Wu and Porté-Agel, 2013). Overall, the results presented here compare favorably to results reported elsewhere in the literature, providing confidence in the physical accuracy of the RANS flow solver.

### 4.2 Layout Optimization for Evenly-Weighted Wind Roses

In order to demonstrate many of the characteristics of optimal layouts and flow fields found using `WindSE`, Fig. 6 shows results obtained for wind roses with an increasing number of evenly-weighted inflow directions. In all cases, the wind speed




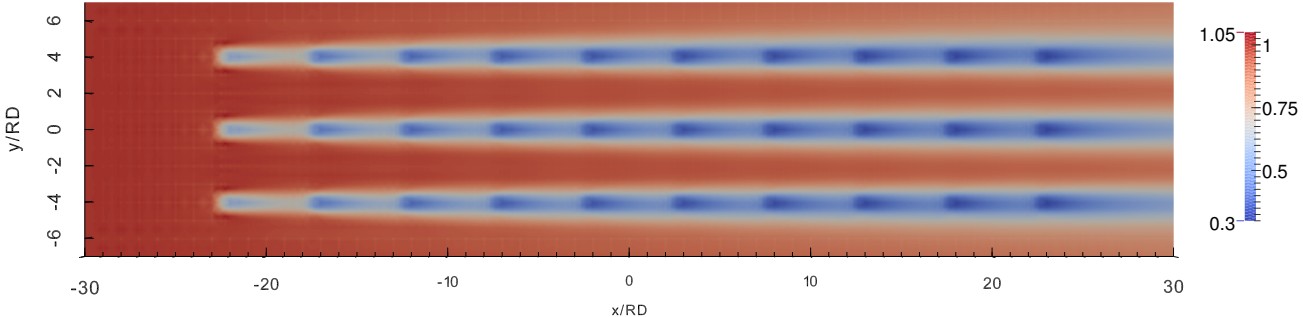

**Figure 5.** Hub-height velocity fields from the RANS solver used in `WindSE` show good qualitative agreement with time-averaged LES velocity fields of similar wind plants reported in the literature (Wu and Porté-Agel, 2013), even in the case of very deep wind plants. Velocities are normalized by incoming hub-height speed of 8 m/s and distances are normalized by the 80 m rotor diameter.

was assumed constant at 8 m/s. During the optimization, the turbines are constrained to stay within an $18\,\mathrm{RD} \times 18\,\mathrm{RD}$ region in the center of the computational domain (see also Fig. 2). This region serves as the site boundary in the present tests. The turbines are initialized on a regular grid that spans $2/3$ of the site constraints (as shown in Fig. 2) and no inter-turbine spacing constraint was applied. Convergence was achieved in approximately $30-50$ iterations of the optimization loop shown in Fig.

3.

Figure 6 shows that optimal layouts are symmetric about a rotation or reflection when the wind directions are evenly weighted. For the two-direction and six-direction cases shown in Figs. 6(a) and (e), the layout is symmetric about a $180°$ rotation and for the four-direction case shown in Fig. 6(c), the layout is symmetric about a $90°$ rotation. For odd numbers of inflow directions, Figs. 6(b) and (d) show that optimal layouts are symmetric about a horizontal reflection across the north-

south axis. The rotational and reflectional symmetries of the layouts for evenly weighted, uniform speed wind roses are useful heuristics for checking the flow solver and optimization results.

The flow fields shown in Fig. 6 indicate that the RANS solver captures flow curvature due to nonlinear turbulent transport effects and pressure increases upwind of the rotor disk. These effects are neglected when superimposing a wake deficit on a flow field obtained from a linear model, as is typically done in many industry-standard optimization frameworks. The flow

curvature results in local speedup effects between two turbines, or just outside a wake, that downwind turbines take advantage of in strongly directional wind roses. This can be observed from the 'staggered' appearance of many of the layouts in Fig. 6 where the optimizer takes advantage of these local speedups. Because the power production scales with the cube of the windspeed, these small speedups can have a strong nonlinear effect on the power output. This speedup effect due to flow curvature is particularly enhanced in 2D, but is still present in the 3D simulations.

Flow curvature also affects the propagation and interaction of the turbine wakes. Wakes near the edge of the plant are slightly deflected away from the plant center and reflect the overall spreading of the flow streamlines. This curvature can be observed near the edges of the plant in the cases shown in Fig. 6. Such curvature is again not captured by engineering wake models





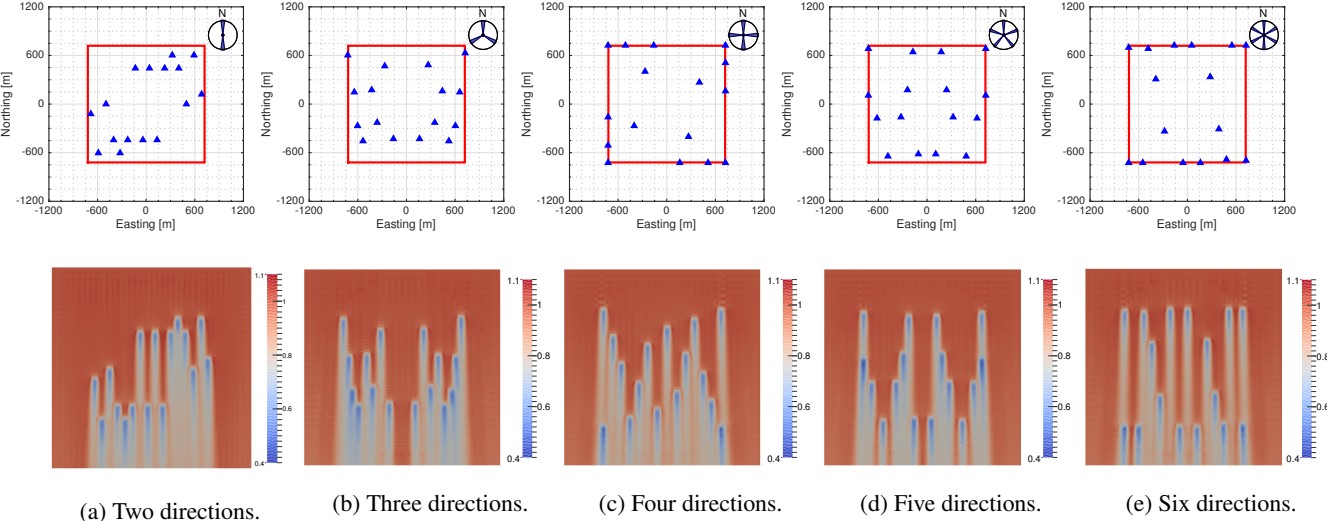

| (a) Two directions. | (b) Three directions. | (c) Four directions. | (d) Five directions. | (e) Six directions. |

**Figure 6.** Optimal layouts (top row) and flow fields (bottom row) for five test cases with an increasing number of evenly-weighted inflow directions. The wind roses in the upper right corner of each layout plot show the inflow directions used in the optimization. The blue triangles show the optimized turbine locations and the red square indicates the site boundaries. The velocities are normalized by the hub-height inflow velocity, which is 8 m/s in these simulations. Each of the flow fields are shown for a wind that blows from the north.

which prescribe wakes that always travel perpendicular to the rotor. Additionally, the wakes are pinched, curved, or merged when encountering speedups around downwind turbines or other wakes. The RANS flow solver accounts for the effects of other turbines and their wakes on the expansion and dissipation of wakes beyond what is accounted for in prescribed wake models. The deflection and curvature of wakes likely has important ramifications for yaw control strategies that attempt to 5 steer wakes away from downwind turbines.

It is emphasized that the results in Fig. 6 show that the optimizer does not place turbines in straight rows perpendicular to a single predominant wind direction or in a regularly spaced grid for these evenly-weighted wind rose cases. Instead, the turbines are placed closer together and staggered to take advantage of local speedups between laterally-placed turbines. This is a different strategy than the maximized spacing found when optimizing with linear flow models and prescribed wakes, and is 10 likely sensitive to the number and size of wind direction bins.

### 4.3 Layout Optimization with Unevenly-Weighted Wind Roses

In the previous section, each inflow direction was given an equal weight (i.e., $\alpha_k$) in creating the total objective function $J$. However, real-world wind roses are seldom so simple and typically have several preferred wind directions, with many other less dominant directions. Here we demonstrate optimization of a 16-turbine layout using the same computational domain and 15 setup as in the previous section, but with unevenly-weighted wind roses that have two and three dominant directions. Once again, in both cases we assume a constant 8 m/s windspeed and do not enforce inter-turbine spacing constraints.





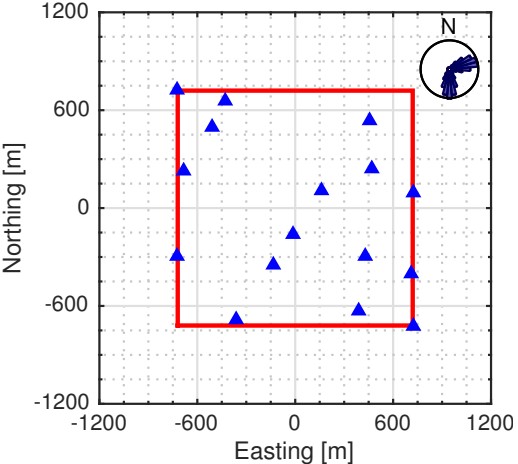
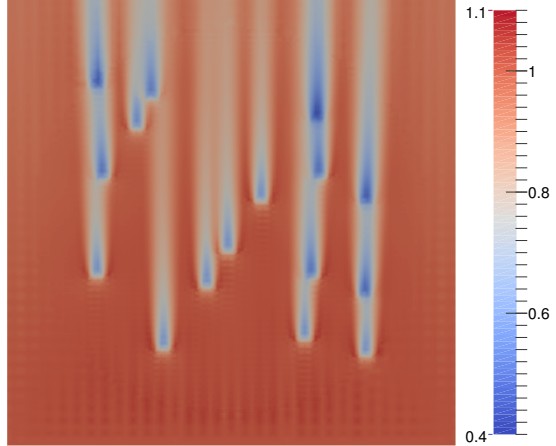

**Figure 7.** Optimal layout (left) and flow field (right) for the boomerang test case with the wind rose binned into 36 inflow directions and a uniform hub-height velocity of 8 m/s. The wind rose is shown in the upper right corner of the left panel, and the site boundary is shown by a solid red line. Turbine locations are denoted by blue triangles. The flow field in the right panel is normalized by the incoming hub-height velocity of 8 m/s and shows results for an inflow wind from the south.

The first wind rose considered has two prominent wind directions spaced $107°$ degrees apart, with a random normal distribution of directions around the primary axes. The wind rose is binned into 36 directions and is roughly shaped like a boomerang, as shown in Fig. 7. Compared to the initial uniform gridded layout (see Fig. 2), the optimized layout shown in Fig. 7 improves power production by 9.4%. Despite the uneven-weighting of the wind rose, the resulting flow field in the right panel of Fig. 7

once again conforms to many of the heuristics outlined in the previous section, including turbines that take advantage of local speedups between upstream turbines and slight flow curvature at the edge of the plant.

The second wind rose considered is given by the directional distribution of 8 m/s wind speeds from the M2 mast at NWTC (Jager, D. and Andreas, A., 1996), as shown in Fig. 8. The wind rose is constructed from publicly available data recorded over the 2015 calendar year. For this windspeed, the wind rose has three prominent directions roughly aligned with inflow from the

10 north, south, and west, along with many other much less prominent inflow directions. The wind rose is again binned into 36 directions (giving $K = 36$) and the optimized layout and resulting flow field are shown in Fig. 8. Compared to the initial regular gridded layout shown in Fig. 2, the power production of the optimized layout is improved by 7.0%. Once again, despite the much more topologically complex nature of the optimized layout shown in Fig. 8, heuristics such as local speed-ups, staggered spacing, and flow curvature are shown to be important in determining the final layout. It is also emphasized that for such a

15 complex wind rose, the resulting layout bears little resemblance to a regular grid where the turbines would be spaced as far from each other as possible.



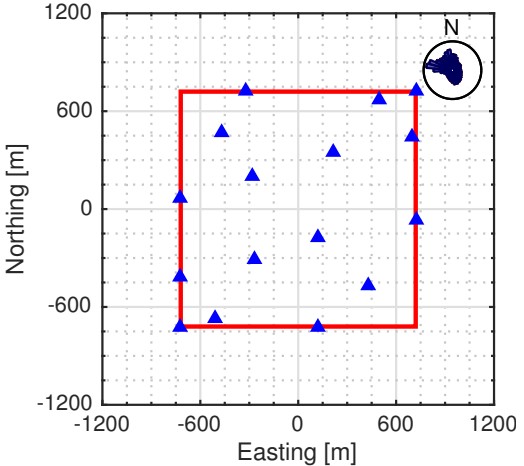 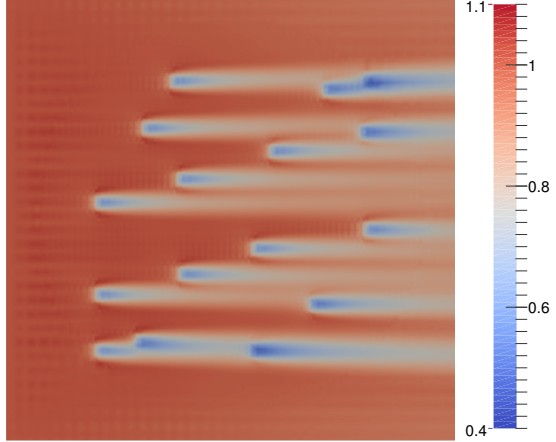

**Figure 8.** Optimal layout (left) and flow field (right) for the NWTC M2 8 m/s test case with the wind rose binned into 36 inflow directions. The wind rose is shown in the upper right corner of the left panel, and the site boundary is shown by a solid red line. Turbine locations are denoted by blue triangles. The flow field in the right panel is normalized by the incoming hub-height velocity of 8 m/s and shows results for an inflow wind from the west.

## 4.4 Layout Optimization Based on Annual Energy Production

As a final demonstration of the power and flexibility of WindSE, we optimize a 16-turbine wind plant based on the full AEP from a real-world site data. In this case, wind states corresponding to both different wind speeds and wind directions are considered, instead of simply considering a single uniform windspeed as in the tests described in the previous two sections.

Data are once again used from the M2 mast at NWTC (Jager, D. and Andreas, A., 1996), and distributions are formed by binning the data into 36 wind directions and 5 windspeed classes centered on 4 m/s, 6 m/s, 8 m/s, 10 m/s, and 12 m/s, giving a total of $K = 180$ possible wind states in the analysis. Wind states that occurred less than 0.2 % of the time were neglected in the optimization, reducing the total number of states considered to 99. We further enforce a minimum inter-turbine spacing $D_{min}$ of three times the rotor diameter.

As shown in Fig. 9, the wind rose for the M2 mast is predominately distributed along the west-northwest direction, with secondary influences from the north and south. The highest wind speeds are also observed for winds from the west-northwest, and so it can be anticipated that a full AEP optimization will result in a layout that is preferentially suited for winds that blow from this direction. This is indeed the case, as shown in the final optimized layout in Fig. 9. The turbines are loosely arranged in two north-south rows that result in relatively large separations between upstream and downstream turbines when

the wind is from the west-northwest. As with other tests for evenly and unevenly weighted wind roses with uniform wind speeds, the turbines in the full AEP optimization are staggered with respect to each other in order to take advantage of local speedups between upwind turbines. The resulting optimized layout improves AEP by 8.6% compared to the initial regular gridded layout shown in Fig. 2.



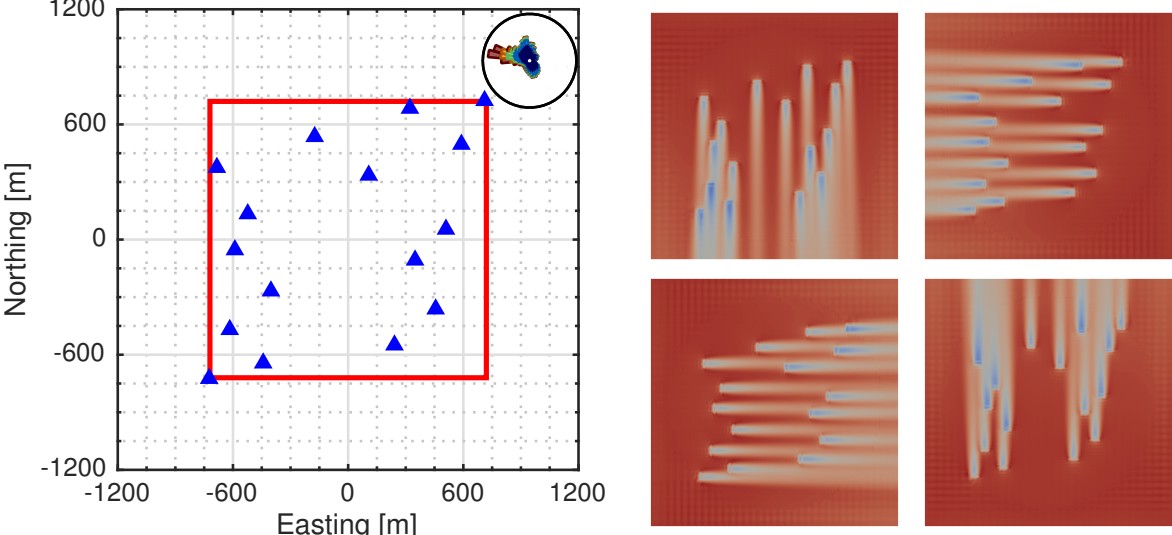

**Figure 9.** Optimal layout (top left) and flow fields resulting from a full AEP optimization using data from the M2 mast at NWTC. Differences in both wind speed and wind direction are accounted for in the optimization, and the wind rose used is shown in the top right corner of the layout plot. In the top left panel, the site boundary is shown by a solid red line and turbine locations are denoted by blue triangles. Flow fields are shown for inflow winds from the west (top right), south (bottom right), and north (bottom left).

It should be noted that despite the predominant high-speed winds from the west-northwest, the turbines in the north-south rows shown in Fig. 9 do not each fall perfectly on the site boundaries. That is, if only winds from the west-northwest were included in the analysis, one might naively place all turbines along the upstream and downstream site boundaries in order to maximize the separation between turbines in the direction of the dominant wind. However, since other wind directions are also included in the analysis, the final optimal layout is more complicated and ensures some degree of turbine staggering in the east-west direction to take advantage of local speedups and minimize wake losses when the wind blows from the north or south. It is also emphasized that simply by accounting for the full AEP in the optimization, the optimal layout in Fig. 9 is substantially different from the optimal layout shown in Fig. 8, where only a single uniform wind speed was considered.

## 5    Summary and Conclusions

`WindSE` represents a fundamentally new approach to utility scale wind plant layout optimization that implements adjoint optimization of turbine locations in a flexible CFD framework. The 3D RANS flow model advances the fidelity of the fluid dynamics in wind plant optimization simulations, but also requires gradient-based optimization techniques because of the expense of solving the RANS PDE constraint. The adjoint optimization framework in `WindSE` provides these gradients, and further enables efficient, high-dimensional optimization of very large control spaces. This enables layout optimization with a first-principles flow model without running expensive LES simulations for tuning purposes. The results presented in this paper



are achieved at a relatively low computational cost as all optimization results were obtained on a single workstation with a 6 core Intel Xeon processor and 32 GB of memory.

The results presented in this paper show that the nonlinear flow effects leading to wake curvature and local speedups are significant when optimizing over a few prominent wind directions. We find consistent rotational symmetry in the optimal
layouts with evenly-weighted inflow directions, suggesting that evenly-weighted wind roses may be useful diagnostic tests for wind plant optimization. As the wind rose is refined into more bin directions, the optimizer is able to take advantage of prominent wind directions and increase energy production. As the number of wind direction and inflow speed combinations increases, `WindSE` is able to perform a full AEP optimization and achieve sizable gains of almost 9% compared to initial gridded layouts. In the full AEP optimization, the optimizer emphasizes the high speed winds from the west-northwest, since
these winds contain the most energy. However, rather than aligning the turbines into rows perpendicular to the incoming wind, the turbines are offset within the general row structure. This is beneficial when the wind is blowing parallel to the row, which is the most common secondary wind direction. It also allows the turbines to take advantage of slight speedup effects around the edges of the wake from upstream turbines. The gradient-based techniques used in this study are inherently constrained to a local search however, and further research is needed to assess what types of initial layouts are needed to draw conclusions
about global optima.

Future work on `WindSE` will explore coupled optimization of turbine locations, hub height, rotor diameter, and control settings. The flow curvature captured in `WindSE` is observed to deflect or modify turbine wakes and will likely have important implications in yaw control optimization applications. Additionally, the automated derivation and discretization of the adjoint equations makes `WindSE` a powerful testbed that can flexibly adapt to new turbulence models or turbine control schemes
without the expense of re-deriving adjoint equations. Finally, the integration of `WindSE` within NREL's systems engineering tools will enable economic analysis and a consideration of LCOE alongside AEP in future optimization studies.

## Acknowledgements

This work was supported by award UGA-0-41026-70 through the Alliance Partner University Program in partnership with the National Renewable Energy Laboratory.



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
