# Peer review of "Optimization of Wind Plant Layouts Using an Adjoint Approach"

_Wind Energy Science, 2016_

## Referee Comment (RC1) · Anonymous Referee #1 · 16 Sep 2016

general comments:

The paper handles the optimization of wind farm layouts using the discrete adjoint approach in CFD (solved via an FEM package in contrast to FVM codes). Instead of LES or 2D RANS, the paper uses 3D RANS in order to achieve a faster, but still reliable optimized layout, which is based on a high-fidelity model (in contrast to analytical wake models). The design parameters are the turbine positions (in contrast to control variables for fixed turbine positions) and the objective function is maximum power and AEP. The paper is very well written. The conception and scope are clear and the topic is of interest.

major specific comments:

section 4.2:

An intuitive guess for an optimized layout for Fig. 6(a) could be a staggered layout

with two rows orthogonal to the wind direction. You say that this results from nonlinear effects, speedup and flow curvature, but it is still surprising, since the wakes are more or less straight. Could there be any numerical issues in the implementation, which leads to this final layout (e.g. too high/low tolerances of the optimizer, negligence of the turbulence model in the discrete adjoints)? Did you run a simulation with a staggered grid from the naïve guess? And is it really worse than the final positioning from the optimization?

minor specific comments:

title:

The title uses the phrase "Adjoint Optimization", although the optimization is gradient-based and the gradients are computed using the adjoint approach (although other authors use a similar phrase, there is no adjoint optimization, since there are adjoints to the optimization are computed). I propose to use a title which considers this difference (e.g. "Optimization of Wind Plant Layouts using the Adjoint Approach").

abstract/section 1:

It should be noted here that you use the discrete adjoint approach.

section 1 et seqq:

It should be mentioned more often that the used RANS flow is steady-state (in contrast to unsteady RANS).

section 2.1:

A note could be made on "...simple terrain with few turbines.", since the paper does not deal with complex terrain, but in principle, the presented tool could be able to handle flows in complex terrain.

section 2.3:

1.) A note should be made that gradient-based optimizations can only find local minima (beside special, convex cases, where a local minimum is the global one). 2.) It is a little misleading that you talk about "backward in time", but steady-state RANS is used later. A small note should be made on that. 3.) "The resulting adjoint gradients are typically more accurate than finite difference gradients". Gradients by finite differences can be of second order, but are the adjoint gradients of second order? If not, the FD could possibly be more accurate.

section 3.1:

Why don't you use a standard turbulence model like k-$\varepsilon$?

section 3.3:

The layout is rotated instead of rotating the boundaries. So after each CFD, do you rotate the layout back for the use in your optimization algorithm? A note on that should be made.

section 3.5:

The gradients of minimum spacing could be derived analytically, couldn't they?

section 4/4.1:

You claim that the "RANS flow solver accurately captures wind turbine wakes", but a verification/validation is not done (see also later: "providing confidence in the physical accuracy of the RANS flow solver"). You could mention that a verification/validation is not in the scope of the paper or you could refer to other publications, where the flow solver setup is verified/validated.

section 4.2:

1.) A graph with the convergence of the optimization could be shown (if there's something interesting to see or in order to show that the optimizer is runs correctly). 2.) Is the Coriolis force included in the flow solver or why there is a curvature near the edges

of the plant? Is it an effect of the domain size?

section 5:

An idea for the future could be a comparison of your optimization setup (including a higher-fidelity model) and an optimization using standard analytical wake models.

technical corrections:

section 3.2:

1.) It should be written, which quantity is shown in Fig. 1(c).

section 3.5:

Shouldn't there be a comma before "if"?

---

## Referee Comment (RC2) · Anonymous Referee #2 · 16 Sep 2016

General Comments

The paper deals with an important subject, e.g. the maximal power output of a wind plant by optimal placing of the turbines. The combination of RANS and adjoint is nowadays mainly used in shape optimization, so this application shows an approach that is not so widely used yet.

The chosen test cases and final case are well suited to build up the understanding of the optimization process, but I found some sketchiness of the analysis. So I would mainly suggest a more careful analysis of the results.

But overall, the paper is well written and for the most part well structured.

Specific Comments

section 2.2

Generally you talk often about something like realistic or fully turbulent flows. But in RANS it's still "just" a snapshot of the averaged flow, perhaps use steady-state sometimes instead of turbulent flow field. Reading this evoked a picture of a fluctuating flow field, which is not true.

second sentence: the information that cheap computational cost allows gradient-free methods implies that gradient methods are suitable for high computational cost. But that's only the case for low amount of design variables or very few methods. A little note on the limits would be nice.

last sentence of first paragraph: Do you expect improvements regarding the found optimum or regarding the computational effort of the optimization process?

section 2.3

generally: how do constraints and the amount of constraints influence the effort of the adjoint approach? As a well-posed optimization problem is crucial, a sentence on this would be nice. This could also be mentioned when introducing the optimization problem in section 3.1, as you could directly refer to your case.

Eq. 5: Usually this equation is given with other derivation (du/dm instead of partial operators, see: Giles and Pierce 2000) operators. Depending on the definition of u and m, this could be used differently. Perhaps add one explanation here.

after eq. 9: Can you give a reason for your statement regarding the accuracy of gradients?

section 3.3

Beginning: From my knowledge of wind farm simulations, the grid dimensions seem to be quite small. Was there a grid study or former set-up validation of the simulations you could refer to?

Is "'do nothing' boundary condition" perhaps better defined by saying "Neumann bound-

ary condition"?

last paragraph: I am not quite sure – also when looking at your flowchart in the next section – when you are calculating your adjoints. Only on base of the averaged flow field or for each of the K wind states? Because if you could compute your complete gradients based on the very last, averaged step, this would mean an extraordinary reduction of complexity and computational effort of this approach.

section 3.4

second sentence after the numbering: you state that less optimization iterations are necessary for multiple inflow states. This is not intuitive on the first look, so perhaps move the sentence about the more convex problem upwards in order to give an explanation.

You mention line search without naming the optimization algorithm you will use. Perhaps move the description (or a shorter version of this part) of SLSQP from the end of section 3.5 to this section.

section 4.1

Plot: The speed-up is very difficult to see on the lower two plots. Perhaps adding the mean velocity in thin lines could be helpful to show where the current velocity is higher or below that value. Another idea would be to mark the areas of increased velocity, for example with shaded backgrounds.

This section is the part where the flow field, that is the base of the optimization, is investigated. Therefore, the statements are a bit weak. Can you either give a quantifying number of the error or cite some publication where this flow solver set-up is validated?

section 4.2

You justify the staggered layout with the speed-ups. But the speed-up only appears very close to the inducing turbine. Or do you talk about something like jets between

two neighbouring turbines? Also in figure 5 this becomes not very clear to me, as the maximal speed-up is less than 5% and only visible right behind the rotor disc. If you talk about two different speed-ups a clear wording has to be found.

Figure 6: The first plot for two wind directions shows a complex layout. I would have guessed either a fully occupied front row of turbines at the north and south boundary of the farm and then some turbines in between with maximal spacing in between in order to reduce the impact of wakes. Or if the speed-up effect is high enough, a pairwise placing of the turbines along the diagonal from lower left to upper right corner of the farm boundaries, because with this layout the turbines either see the full wind or the speed-up wind from the one in front. Did you check these layouts? If so, how do they perform? How do the turbines move? Because the shown displacement of some turbines seems to be quite big, what is a bit surprising for a local optimum search. Perhaps a movement trajectory of some representative turbines helps to understand the final layouts. Discussing this most easy case more in detail also helps to understand and analyse the other results.

Technical Comments

Figure 2: Axis labels and numbers are quite small

In the figures where you show velocity fields it can be helpful to reduce the amount of colour steps (eg. between red and blue). That way differences (and speed-ups) can become more obvious.
* * *

---

## Referee Comment (RC3) · Anonymous Referee #3 · 12 Oct 2016

The authors present work on wind-farm layout optimization using RANS and gradient-based optimization using adjoints for the determination of the gradients. The work is interesting and merits publications. Nevertheless, I have a number of (relatively minor) comments that should be addressed first.

1. Page 5, Eq 1: formally the state u should be added under the min, so $\min_{m,u}$ (in this formulation, optimization is done over state and control, given the state and other constraints]

2. page 6: the formalism for the derivation of the adjoint should be cleaned up a bit. First of all, there is a distinction between $J(u, m)$ and $\hat{J}(m) \equiv J(u(m), m)$, where $u(m)$ is defined by $F(u(m), m) \equiv 0$. The authors want to derive the gradient of $\hat{J}(m)$ (to m), not the gradient of $J$ to m, which is simply $\partial J/\partial m$. In fact, the authors do not solve (1)-(4) directly, but rather a reduced formulation, i.e. $\min_m \hat{J}(m)$ (s.t. to all constraints except the state constraint, which is now already implicitly containted in $\hat{J}(m)$). This is

a quite common approach in PDE-constrained optimization (cf. e.g. the book by Borzi and Schultz, Computational Optimization of Systems Governed by Partial Differential Equations). Please clarify your notations accordingly in the manuscript

3. page 7: optimization problem should also include the boundary conditions + explicitly express that the turbine forces are function of $m$. Also similar to above, this formulation requires $\min_{m,u}$

4. page 9: I find it very weird to talk about a smoothed thrust coefficient. This goes really against the normal definition of a thrust coefficient, which is a scalar value. Instead, please use the convention that the force is smoothed out over the RANS grid using a geometric smoothing function. This is the standard convention used in all Actuator Disk representations in literature (both RANS and LES – check any of the relevant papers)

5. page 10: given the domain size (height and width), please provide and discus the blockage ratio

6. page 12 , point 6: what optimization method is used?

7. page 14, line 5. Please discuss in more detail what optimization method is used to solve the QP problems: Newton, quasi-Newton, what precise method (truncated, BFGS, thrust-region, . . .)

8. page 14, and results section: you claim the use SQP, but then in the results section, you seem to mention that you do not include the distance constraint. What is the point then in using SQP? Please elaborate. Why did the distance constraint not work? And if not included, why not use a simple box-constrained quasi-Newton method?

9. page 15, figure 4 and discussion: Reference data (experiments or LES) should be added to the plot (in particular to Fig 4b and c). The porous disk is well documented experimentally as well as numerically (and has recently also been used in an inter-comparison study Lignarolo 2016) – without adding reference data, the later statement

'Overall, the results presented here compare favorably to results reported elsewhere' is not verifiable. Given that the turbulence model used is really simple, there might be some differences with profiles from literature (which in itself is not a problem given that the authors develop a new approach) – this has to be properly discussed

10. Section 4.2: It would be interesting to also add a single wind direction + discussion. Even though such a case may not occur in reality, I believe it can yield extra theoretical insight

11. Section 4.2: please provide relative efficiencies of the optimized layouts (compared to all inflow turbines).

12. Section 4.4: Simulations over different wind speeds should not be necessary (since, if I'm not mistaken, you work with a constant thrust coefficient per turbine, so you perform optimization for region II). Your simulations when normalized with wind speed, are independent of wind speed, so relative power doesn't change. Therefore, per wind direction, only wind-speed is necessary. Power will then just scale with cube of wind speed. Please reconsider approach in section 4.4 accordingly

13. page 20 (and other locations earlier): the effect of flow-curvature is maybe a bit too much emphasized. It is certainly true that the current RANS approach allows flow curvature (e.g. not present in the Jensen model), but the authors do not substantiate the claim that flow curvature is an essential feature for optimal layout (i.e. an effect that makes a significant difference). Either substantiate (eg. Based on detailed comparisons with other models that do not have the effect) or tune down the statement.

---

## Author Comment (AC1) · 10 Nov 2016

**Response to Reviewer #1**

We appreciate the constructive and insightful comments from the reviewer. Detailed replies to each of the reviewer's points (in blue italics) are provided below. Given the opportunity, we feel that incorporating the reviewer's comments will produce a substantially clearer and stronger paper as a result.

*general comments:*

*The paper handles the optimization of wind farm layouts using the discrete adjoint approach in CFD (solved via an FEM package in contrast to FVM codes). Instead of LES or 2D RANS, the paper uses 3D RANS in order to achieve a faster, but still reliable optimized layout, which is based on a high-fidelity model (in contrast to analytical wake models). The design parameters are the turbine positions (in contrast to control variables for fixed turbine positions) and the objective function is maximum power and AEP. The paper is very well written. The conception and scope are clear and the topic is of interest.*

*major specific comments:*

*section 4.2:*

*An intuitive guess for an optimized layout for Fig. 6(a) could be a staggered layout with two rows orthogonal to the wind direction. You say that this results from nonlinear effects, speedup and flow curvature, but it is still surprising, since the wakes are more or less straight. Could there be any numerical issues in the implementation, which leads to this final layout (e.g. too high/low tolerances of the optimizer, negligence of the turbulence model in the discrete adjoints)? Did you run a simulation with a staggered grid from the naïve guess? And is it really worse than the final positioning from the optimization?*

In response to this comment and suggestions by other reviewers, we will include a comparison of the optimization results to a baseline case with turbines arranged in two rows orthogonal to the wind direction. We also plan to better demonstrate the curvature effects by including a figure with spanwise velocity, such as the figure below showing the spanwise velocity for the test case in Figure 6d. The adjoint equations are discretized in the same manner as the forward problem and have the same solution accuracy. There is no frozen turbulence assumption in our formulation.

[Figure]

*minor specific comments:*

*title:*

*The title uses the phrase "Adjoint Optimization", although the optimization is gradient-based and the gradients are computed using the adjoint approach (although other authors use a similar phrase, there is no adjoint optimization, since there are adjoints to the optimization are computed). I propose to use a title which considers this difference (e.g. "Optimization of Wind Plant Layouts using the Adjoint Approach").*

We appreciate the reviewer's nuanced terminology. We will change the title to "Optimization of Wind Plant Layouts Using an Adjoint Approach."

*abstract/section 1:*

*It should be noted here that you use the discrete adjoint approach.*

We discuss the discrete nature of our adjoint solutions in Section 3.5, and will add a mention of this at the beginning of the paper.

*section 1 et seqq:*

*It should be mentioned more often that the used RANS flow is steady-state (in contrast to unsteady RANS).*

We agree with the reviewer that it is important to convey that our simulations are steady-state, and we will include additional mentions of this point in the abstract, introduction, and conclusions.

*section 2.1:*

*A note could be made on "...simple terrain with few turbines," since the paper does not deal with complex terrain, but in principle, the presented tool could be able to handle flows in complex terrain.*

This is an important point and we will explicitly point out in the methodology section and conclusions that the FEM approach and 3D flow solver are capable of handling complex terrain in future studies.

*section 2.3:*

*1.) A note should be made that gradient-based optimizations can only find local minima (beside special, convex cases, where a local minimum is the global one).*

We agree that this is an important point and we have emphasized it in a number of places in the manuscript. In particular, we discuss local minima in the Background section on line 1 page 4, provide a paragraph on the local nature of our solutions in the Methodology section on lines 6-15, page 12, and readers are further reminded of the local nature in our Summary and Conclusions section on lines 12 - 15, page 21.

*2.) It is a little misleading that you talk about "backward in time", but steady-state RANS is used later. A small note should be made on that.*

In order to avoid confusion, we will add a clarifying comment that the "backward in time" discussion is intended to help readers develop a intuition for how the adjoint approach works in general, but that our specific study considers only steady state simulations where the adjoint variable is also steady-state.

*3.) "The resulting adjoint gradients are typically more accurate than finite difference gradients". Gradients by finite differences can be of second order, but are the adjoint gradients of second order? If not, the FD could possibly be more accurate.*

Since this is, as the reviewer correctly points out, a conditional statement depending on the specific details of how the finite difference and adjoint gradients are calculated, we will remove this statement from the manuscript in order to avoid confusion.

*section 3.1:*

*Why don't you use a standard turbulence model like $k - \epsilon$?*

We chose a mixing length turbulence model over a $k-\epsilon$ model because the mixing length model avoids the use of limiters and min/max functions that are non-differentiable and maintains a simple mathematical presentation. Mixing length models are familiar to the wind energy community through their use in eddy viscosity wake models [e.g., Ainslie (1988)] and in recent RANS models developed for wind plant controls [e.g., Boersma et al., (TORQUE 2016)]. The mixing length model results compare reasonably well to time averaged LES model results while maintaining a simple and optimization-friendly mathematical formulation. There are many improvements to the governing equations that could be made in future studies (more sophisticated turbulence models, buoyancy, Coriolis forces) but were beyond the scope of this paper as our focus was on the gradient-based optimization framework and adjoint techniques. In order to make our selection of turbulence model more clear, we will add a statement to the manuscript clarifying our decision.

*section 3.3:*

*The layout is rotated instead of rotating the boundaries. So after each CFD, do you rotate the layout back for the use in your optimization algorithm? A note on that should be made.*

The control variables that store the turbine positions are kept in a fixed reference frame, and are updated at each optimization iteration. To compute the turbine force corresponding to each inflow direction, we rotate the layout when calculating $f_k$. This rotation is a simple analytical function that is a part of calculating the turbine body force and is incorporated into the adjoint gradients. In order to clarify this, we will modify Eq. (13) to make the dependence of the turbine body force on the inflow angle and reference frame positions explicit.

*section 3.5:*

*The gradients of minimum spacing could be derived analytically, couldn't they?*

Yes, they are derived analytically in our implementation. We intended to convey this by stating that the gradients of the spacing constraint are "provided manually" in line 9, page 14, but will reword it to clarify that we mean analytically.

*section 4/4.1:*

*You claim that the "RANS flow solver accurately captures wind turbine wakes", but a verification/validation is not done (see also later: "providing confidence in the physical accuracy of the RANS flow solver"). You could mention that a verification/validation is not in the scope of the paper or you could refer to other publications, where the flow solver setup is verified/validated.*

We agree with the reviewer that a detailed verification/validation is beyond the scope of this paper, which is focused on the formulation of an optimization and flow solver framework that can be used with a wide variety of different RANS models. We will reiterate this point in the introduction and also tone down our statements of accuracy by noting that the RANS flow solver "qualitatively" agrees with prior studies.

*section 4.2:*

*1.) A graph with the convergence of the optimization could be shown (if there's something interesting to see or in order to show that the optimizer is runs correctly).*

We considered adding just such a plot, but found it relatively uninteresting and not worth including in the submitted manuscript. Nevertheless, we have provided here for the reviewer a plot of the objective function convergence for the two direction optimization case described in Section 4.3 that is representative of all the optimizations.

[Figure]

*2.) Is the Coriolis force included in the flow solver or why there is a curvature near the edges of the plant? Is it an effect of the domain size?*

A Coriolis force is not included in our governing equations. The wakes deflect away from the center of the plant due to slowing of the flow and spreading of the streamlines caused by the continuity equation. We will make this more clear by including a figure with streamlines that will show this effect.

*section 5:*

*An idea for the future could be a comparison of your optimization setup (including a higher-fidelity*

*model) and an optimization using standard analytical wake models.*

We are working on exactly this topic for a followup paper and will make mention of it in our future work.

*technical corrections:*

*section 3.2:*

*1.) It should be written, which quantity is shown in Fig. 1(c).*

We will clarify this in the figure caption.

*section 3.5:*

*Shouldn't there be a comma before "if"?*

Thank you for catching this, and we will correct the sentence.

Sincerely, the authors.

---

## Author Comment (AC2) · 10 Nov 2016

**Response to Reviewer #2**

We appreciate the constructive and insightful comments from the reviewer. Detailed replies to each of the reviewer's points (in blue italics) are provided below. Given the opportunity, we feel that incorporating the reviewer's comments will produce a substantially clearer and stronger paper as a result.

*General Comments*

*The paper deals with an important subject, e.g. the maximal power output of a wind plant by optimal placing of the turbines. The combination of RANS and adjoint is nowadays mainly used in shape optimization, so this application shows an approach that is not so widely used yet.*

*The chosen test cases and final case are well suited to build up the understanding of the optimization process, but I found some sketchiness of the analysis. So I would mainly suggest a more careful analysis of the results.*

*But overall, the paper is well written and for the most part well structured.*

*Specific Comments*

*section 2.2*

*Generally you talk often about something like realistic or fully turbulent flows. But in RANS it's still "just" a snapshot of the averaged flow, perhaps use steady-state sometimes instead of turbulent flow field. Reading this evoked a picture of a fluctuating flow field, which is not true.*

> We agree with the reviewer that it is important to convey that our simulations are steady-state, and we will provide additional references to this fact in the abstract, introduction, and conclusions.

*second sentence: the information that cheap computational cost allows gradient-free methods implies that gradient methods are suitable for high computational cost. But that's only the case for low amount of design variables or very few methods. A little note on the limits would be nice.*

> We agree that gradient-based methods are suitable for optimizing high computation cost functionals because minima can be found with orders of magnitude fewer functional evaluations than a gradient-free method. It is also true that certain gradient methods may be limited to a relatively small number of control variables if the cost of calculating the gradient scales with the number of control variables, such as in a finite difference approach. However, a core point of our paper is that the adjoint approach finds gradients at a cost that is independent of the number of control variables, allowing for gradient-based optimization of high-dimensional optimization problems with many controls variables. In order to avoid confusion, we will more clearly make a distinction in this section between other gradient-based methods and the present adjoint approach to finding gradients.

*last sentence of first paragraph: Do you expect improvements regarding the found optimum or regarding the computational effort of the optimization process?*

> We expect that using a higher fidelity flow model would result in a different optimal layout than optimizing with a linear wake model because the optimization algorithm would see different representations of the wake and underlying flow physics. We are preparing a follow-up paper

comparing optimized layouts obtained from our model and an engineering wake model when both layouts are simulated in an LES model. We will make mention of this in our future work plans at the end of the paper.

*section 2.3*

*generally: how do constraints and the amount of constraints influence the effort of the adjoint approach? As a well-posed optimization problem is crucial, a sentence on this would be nice. This could also be mentioned when introducing the optimization problem in section 3.1, as you could directly refer to your case.*

The most challenging constraints are related to the inter-turbine minimum spacing as the number of these constraints grows exponentially and requires the use of the SLSQP optimization algorithm instead of a bounded BFGS algorithm. Following the reviewer's recommendation, we will expand our discussion of the constraints and their impact on selecting an optimization algorithm in Sections 2.3 and 3.1

*Eq. 5: Usually this equation is given with other derivation (du/dm instead of partial operators, see: Giles and Pierce 2000) operators. Depending on the definition of u and m, this could be used differently. Perhaps add one explanation here.*

Based on this comment and a comment from another reviewer we will reformulate the adjoint derivation in terms of reduced functionals that should avoid any confusion.

*after eq. 9: Can you give a reason for your statement regarding the accuracy of gradients?*

Given that the accuracy of the gradients depends heavily on the specific calculation method (e.g., 2nd vs. 4th order finite differences), we have chosen to remove this sentence in order to avoid ambiguity.

*section 3.3*

*Beginning: From my knowledge of wind farm simulations, the grid dimensions seem to be quite small. Was there a grid study or former set-up validation of the simulations you could refer to?*

We agree that the grid dimensions in this study could be increased, but our primary interest here is in introducing the adjoint optimization and flow solver tool. For this we chose a more 'manageable' grid size in order to explore a wider range of cases. In the future, we will implement and test various improvements to the tool, including more sophisticated RANS models, complex terrain, and more grid points. We will expand our discussion of future work in the conclusions to point out that larger problem sizes are an important area of development.

*Is " 'do nothing' boundary condition" perhaps better defined by saying "Neumann boundary condition"?*

The 'do nothing' boundary condition is the typical outflow boundary condition in finite element simulations of incompressible flows where the boundary integral terms arising from integration by parts of the variational form are set to 0, i.e., $\nu \partial_n u - pn = 0$ on the outflow boundary. For readers seeking more detail on this boundary condition, we will add a reference to Heywood, Rannacher, and Turek (Int. J. of Num. Meth. in Fluids, 1996).

 *I am not quite sure when looking at your flowchart in the next section when you are calculating your adjoints. Only on base of the averaged flow field or for each of the K wind states? Because if you could compute your complete gradients based on the very last, averaged step, this would mean an extraordinary reduction of complexity and computational effort of this approach.*

The adjoint is calculated after solving for each of the flow states and calculating a total power. This total summation involves the flow field solutions from each state, and as a result a corresponding adjoint simulation is required for each one. In order to make this more clear we will add a sentence further explaining this point.

*section 3.4*

*second sentence after the numbering: you state that less optimization iterations are necessary for multiple inflow states. This is not intuitive on the first look, so perhaps move the sentence about the more convex problem upwards in order to give an explanation.*

We will reorder the sentences accordingly.

*You mention line search without naming the optimization algorithm you will use. Perhaps move the description (or a shorter version of this part) of SLSQP from the end of section 3.5 to this section.*

We appreciate this suggestion and we will move the discussion of optimization algorithms from Section 3.5 up to Section 3.4.

*section 4.1*

*Plot: The speed-up is very difficult to see on the lower two plots. Perhaps adding the mean velocity in thin lines could be helpful to show where the current velocity is higher or below that value. Another idea would be to mark the areas of increased velocity, for example with shaded backgrounds.*

In order to clarify this figure, we will reduce the number of profiles shown in the bottom two panels and will also add vertical dashed lines to each profile showing the baseline value of velocity for comparison. We also wish to point out that this figure was not intended to demonstrate the speedup so much as to demonstrate correct wake and boundary layer profiles. As the reviewer correctly notes in a later comment, the speedup effects are more apparent between two adjacent turbines than around just a single turbine.

*This section is the part where the flow field, that is the base of the optimization, is investigated. Therefore, the statements are a bit weak. Can you either give a quantifying number of the error or cite some publication where this flow solver set-up is validated?*

In this section our intent was to show that the flow fields produced are qualitatively consistent with prior results. Moreover, the primary purpose of this paper is to introduce the combined adjoint and flow solver optimization tool. In reality, a wide range of different RANS models could be used with this tool, and exploring different models is a direction for future work. In order to make the purpose of this section more clear, we will note at the beginning and end of the section that the purpose of these flow solver tests is simply to demonstrate qualitative agreement with prior results and that a wide range of other flow solvers and RANS models could be implemented in the tool.

*You justify the staggered layout with the speed-ups. But the speed-up only appears very close to the inducing turbine. Or do you talk about something like jets between two neighbouring turbines? Also in figure 5 this becomes not very clear to me, as the maximal speed-up is less than 5% and only visible right behind the rotor disc. If you talk about two different speed-ups a clear wording has to be found.*

The reviewer is correct that the speedup effect we refer to might be better described as a jet between two turbines. The speedup effect is enhanced by the presence of two turbines and is less noticeable for a single turbine in isolation. We will clarify our description of the speedup effect accordingly. The power produced by the turbines scales with the cube of the windspeed, so even slight windspeed increases can produce substantial power increases.

*Figure 6: The first plot for two wind directions shows a complex layout. I would have guessed either a fully occupied front row of turbines at the north and south boundary of the farm and then some turbines in between with maximal spacing in between in order to reduce the impact of wakes. Or if the speed-up effect is high enough, a pair-wise placing of the turbines along the diagonal from lower left to upper right corner of the farm boundaries, because with this layout the turbines either see the full wind or the speed-up wind from the one in front. Did you check these layouts? If so, how do they perform? How do the turbines move? Because the shown displacement of some turbines seems to be quite big, what is a bit surprising for a local optimum search. Perhaps a movement trajectory of some representative turbines helps to understand the final layouts. Discussing this most easy case more in detail also helps to understand and analyse the other results.*

In response to this comment and one from another reviewer, we will add a reference case with two parallel turbine rows and discuss improvements with the optimal layout.

*Technical Comments*

*Figure 2: Axis labels and numbers are quite small*

We will increase the font size.

*In the figures where you show velocity fields it can be helpful to reduce the amount of colour steps (eg. between red and blue). That way differences (and speed-ups) can become more obvious.*

We appreciate this suggestions and we will try reducing the number of discrete colors to see if the effects become more apparent.

Sincerely, the authors.

---

## Author Comment (AC3) · 10 Nov 2016

**Response to Reviewer #3**

We appreciate the constructive and insightful comments from the reviewer. Detailed replies to each of the reviewer's points (in blue italics) are provided below. Given the opportunity, we feel that incorporating the reviewer's comments will produce a substantially clearer and stronger paper as a result.

*The authors present work on wind-farm layout optimization using RANS and gradient-based optimization using adjoints for the determination of the gradients. The work is interesting and merits publications. Nevertheless, I have a number of (relatively minor) comments that should be addressed first.*

*1. Page 5, Eq 1: formally the state u should be added under the min, so $\min_{m,u}$ (in this formulation, optimization is done over state and control, given the state and other constraints).*

> We agree, as explained more fully in the next response.

*2. page 6: the formalism for the derivation of the adjoint should be cleaned up a bit. First of all, there is a distinction between $J(u,m)$ and $\hat{J}(m) \equiv J(u(m),m)$, where $u(m)$ is defined by $F(u(m),m) \equiv 0$. The authors want to derive the gradient of $\hat{J}(m)$ (to m), not the gradient of J to m, which is simply $\partial J/\partial m$. In fact, the authors do not solve (1)-(4) directly, but rather a reduced formulation, i.e. $\min_m \hat{J}(m)$ (s.t. to all constraints except the state constraint, which is now already implicitly contained in $\hat{J}(m)$. This is a quite common approach in PDE-constrained optimization (cf. e.g. the book by Borzi and Schultz, Computational Optimization of Systems Governed by Partial Differential Equations). Please clarify your notations accordingly in the manuscript*

> We appreciate the reviewer's suggestions concerning the optimization variables and use of a reduced functional. We will modify our formulation to include these changes, resulting in a cleaner description and additional rigor in the optimization formulation.

*3. page 7: optimization problem should also include the boundary conditions + explicitly express that the turbine forces are function of m. Also similar to above, this formulation requires $\min_{m,u}$*

> Our boundary conditions are currently discussed in Section 3.3 and we agree that they should also be included in the optimization problem formulation. In particular, we will include the boundary conditions as well as make explicit the dependence of the turbine forces on $m$.

*4. page 9: I find it very weird to talk about a smoothed thrust coefficient. This goes really against the normal definition of a thrust coefficient, which is a scalar value. Instead, please use the convention that the force is smoothed out over the RANS grid using a geometric smoothing function. This is the standard convention used in all Actuator Disk representations in literature (both RANS and LES – check any of the relevant papers)*

> We are in full agreement with the reviewer and we do in fact treat the power and thrust coefficients as scalars, indicated by the variable $c'$. Moreover, our smoothing kernel in Eq. 23 is exactly the geometric smoothing function suggested by the reviewer. Our smoothed field $C'$ was introduced to simplify the objective function expression. In order to avoid confusion on this point and clarify the presentation in our revision, we will replace $C'$ with the explicit use of the power and thrust coefficients and geometric smoothing kernel.

*5. page 10: given the domain size (height and width), please provide and discuss the blockage ratio*

In wind tunnel studies the blockage ratio is typically the ratio of the total tunnel cross-sectional area to the total rotor disk area. The total frontal area of our simulation domain is $8RD \times 30RD = 1.536 \times 10^6$ m$^2$ for our turbines with 80 m rotor diameters. Our starting layouts contain a $4 \times 4$ grid of turbines, resulting in $4 \times \pi \times 40^2 = 2.01 \times 10^4$ m$^2$, or $1.3\%$ blockage effective blockage. A worst case scenario would have all 16 turbines arranged in a line perpendicular to the inflow direction, resulting in $5.2\%$ blockage. We believe this blockage is sufficiently small to avoid requiring any correction effects based on a recent study which confirmed these corrections were only necessary above blockage ratios of $10\%$ (Chen and Liou, Experimental Thermal and Fluid Sciences, 2011). We will include the blockage ratio and reference information in our domain description.

*6. page 12 , point 6: what optimization method is used?*

Please see next response.

*7. page 14, line 5. Please discuss in more detail what optimization method is used to solve the QP problems: Newton, quasi-Newton, what precise method (truncated, BFGS, thrust-region, ...)*

We have used SLSQP when we enforce an inter-turbine spacing constraint and L-BFGS-B when only enforcing site boundaries and not spacing constraints. The initial optimization studies we present are intended to provide the reader with an intuition about the flow physics that the optimizer uses, and consequently we used L-BFGS-B without inter-turbine spacings. The later annual energy production optimization results we present are intended to focus on more real-world optimization, and there we do enforce inter-turbine spacing constraints and switch to the SLSQP optimizer. We will clarify our selection of optimization algorithms in Section 3.4.

*8. page 14, and results section: you claim the use SQP, but then in the results section, you seem to mention that you do not include the distance constraint. What is the point then in using SQP? Please elaborate. Why did the distance constraint not work? And if not included, why not use a simple box-constrained quasi-Newton method?*

For all of the real-world wind rose results in Section 4.4 we imposed the turbine spacing constraint and used the SLSQP algorithm to optimize over the full annual energy production. The optimizations presented in Sections 4.2 and 4.3 were simplified test cases with idealized wind roses and no spacing constraints to help demonstrate the flow physics that produced optimal layouts. These simplified constraints allowed us to use the L-BFGS-B algorithm. We will clarify our selection of optimization algorithms in Section 3.4.

*9. page 15, figure 4 and discussion: Reference data (experiments or LES) should be added to the plot (in particular to Fig 4b and c). The porous disk is well documented experimentally as well as numerically (and has recently also been used in an intercomparison study Lignarolo 2016); without adding reference data, the later statement "Overall, the results presented here compare favorably to results reported elsewhere" is not verifiable. Given that the turbulence model used is really simple, there might be some differences with profiles from literature (which in itself is not a problem given that the authors develop a new approach) – this has to be properly discussed*

The reviewer is correct that, in this paper, we simply seek to introduce the coupled adjoint and flow solver optimization approach. As such, exact or quantitative flow solver agreement with prior results is not strictly required, and is likely to be strongly dependent on the choice of RANS flow model. Our intent in Section 4.1 was simply to provide confidence that the present flow solver results are *qualitatively* consistent with prior studies. In order to make this more clear, we will emphasize at the beginning and end of this section that the objective is to show that the present flow solver is a reasonable choice for introduction of the tool, and that more sophisticated RANS models or flow solvers can be implemented in the future.

*10. Section 4.2: It would be interesting to also add a single wind direction + discussion. Even though such a case may not occur in reality, I believe it can yield extra theoretical insight*

We will add a single inflow direction optimization to Figure 6.

*11. Section 4.2: please provide relative efficiencies of the optimized layouts (compared to all inflow turbines).*

We will provide such a calculation in our revision.

*12. Section 4.4: Simulations over different wind speeds should not be necessary (since, if I'm not mistaken, you work with a constant thrust coefficient per turbine, so you perform optimization for region II). Your simulations when normalized with wind speed, are independent of wind speed, so relative power doesn't change. Therefore, per wind direction, only wind-speed is necessary. Power will then just scale with cube of wind speed. Please reconsider approach in section 4.4 accordingly*

We in fact had a similar idea during preparation of this manuscript and compared cases with and without constant wind speeds. From these tests, we found that the simulations actually do depend on wind speed, even when the thrust coefficient is constant. Our explanation for this is that, even when normalizing by the inflow hub height velocity, the simulation Reynolds number will change, and therefore so will the wake spreading and dissipation rates. We are happy to provide a comment regarding this point to the revised manuscript.

*13. page 20 (and other locations earlier): the effect of flow-curvature is maybe a bit too much emphasized. It is certainly true that the current RANS approach allows flow curvature (e.g. not present in the Jensen model), but the authors do not substantiate the claim that flow curvature is an essential feature for optimal layout (i.e. an effect that makes a significant difference). Either substantiate (e.g. Based on detailed comparisons with other models that do not have the effect) or tune down the statement.*

Our optimized layouts show that turbines behind the first row are placed in locations where streamlines are converging and flow is accelerating between turbines. Flow curvature and the resulting acceleration between turbines is the mechanism that underlies this optimization heuristic, which is not present in, e.g., the Jensen model. In order to emphasize the significance of the curvature effect in turbine placement, we will provide a new figure that shows the spanwise velocities (which are not present in a Jensen wake model) to demonstrate how the optimal turbine placement is driven by curvature and the resulting speedups. An example figure with the spanwise velocities for the layout in Figure 6d is shown below.

[Figure]

Sincerely, the authors.

---

## Author Response (AR1)

We appreciate the opportunity to revise our manuscript and incorporate feedback from the referees. A revised version of our manuscript is included below with changes marked in red. Additionally, we have included a short summary of the most significant changes to the text.

**Response to Editor's Comments**

*Dear authors,*

*Please perform the mentioned revisions, please make your changes clear to me, so that I can easily see if the paper can be finally accepted.*

Please see our revised manuscript below with changes marked in red. The major changes that we have made in response to the reviewer comments are:

1. A rewording of the manuscript title to avoid confusion about the use of adjoint techniques.

2. Increased emphasis on the steady-state nature of our RANS solver, its qualitative performance, and appropriateness of the mixing length closure to demonstrate the adjoint optimization framework.

3. A revised notation for the statement of our optimization problem that makes explicit the use of a reduced functional and dependencies on the control variables.

4. A clarification of the notation regarding the geometric smoothing kernel used in our actuator disk.

5. An additional note on the blockage ratio of our simulations.

6. A clarification of how the turbine coordinates are defined and how the turbine layout is rotated to simulate each inflow direction.

7. A clarification on the choice of optimization algorithms and implementation of inter-turbine spacing constraints.

8. A new figure showing spanwise velocities that demonstrates flow curvature and differentiates our RANS model results from linear wake model results.

9. A comparison of a 'naïve' layout containing two parallel turbine rows to our optimized layout for the two inflow direction test case, further demonstrating the gains from optimization.

10. A new table describing the relative performance of the optimized layouts as compared to unwaked turbines.

Sincerely, the authors.

[revised manuscript text omitted]